# Ordering-based Conditions for Global Convergence of Policy Gradient Methods

**Jincheng Mei**
Google DeepMind
jcmei@google.com

**Bo Dai**
Google DeepMind
bodai@google.com

**Alekh Agarwal**
Google Research
alekhagarwal@google.com

**Mohammad Ghavamzadeh**
Amazon*
ghavamza@amazon.com

**Csaba Szepesvári**
Google DeepMind
University of Alberta
szepi@google.com

**Dale Schuurmans**
Google DeepMind
University of Alberta
daes@ualberta.ca

## Abstract

We prove that, for finite-arm bandits with linear function approximation, the global convergence of policy gradient (PG) methods depends on inter-related properties between the policy update and the representation. First, we establish a few key observations that frame the study: **(i)** Global convergence can be achieved under linear function approximation without policy or reward realizability, both for the standard Softmax PG and natural policy gradient (NPG). **(ii)** Approximation error is not a key quantity for characterizing global convergence in either algorithm. **(iii)** The conditions on the representation that imply global convergence are different between these two algorithms. Overall, these observations call into question approximation error as an appropriate quantity for characterizing the global convergence of PG methods under linear function approximation. Second, motivated by these observations, we establish new general results: **(i)** NPG with linear function approximation achieves global convergence *if and only if* the projection of the reward onto the representable space preserves the optimal action's rank, a quantity that is not strongly related to approximation error. **(ii)** The global convergence of Softmax PG occurs if the representation satisfies a non-domination condition and can preserve the ranking of rewards, which goes well beyond policy or reward realizability. We provide experimental results to support these theoretical findings.

## 1 Introduction

Policy gradient (PG) is a foundational concept in reinforcement learning (RL), centrally used in both policy-based and actor-critic methods [25]. Despite the non-convexity of the policy optimization objective [4], global convergence of PG methods has been recently established in the tabular case for standard configurations such as the softmax parameterization [4, 22] and stochastic on-policy sampling [20]. In practice, when an RL agent is faced with a problem with large state and/or action spaces, *function approximation* is needed to generalize across related states and actions. The behavior of PG methods in these settings is relatively under-explored. In this paper, we study this question for the case of linear function approximation, and establish a surprising result that

*the classical Softmax PG method converges whenever there exists an adequate linear function that ranks actions in the same order as the ground-truth reward function.*

---

*The work was done prior to joining Amazon, while the author was at Google Research.

37th Conference on Neural Information Processing Systems (NeurIPS 2023).

Understanding the behavior of PG methods under function approximation is crucial for describing the behavior of RL in practice, since one rarely faces domains small enough to explicitly enumerate over states and actions in parameterizing the policy. It is well known that, standard Softmax PG converges to stationary points if a "compatible" function approximation is used [25]; i.e., one that is able to exactly represent policy value functions. However, when exact policy values are non-realizable, "approximation error" is typically considered to be the key quantity for characterizing how well a function approximation captures relevant problem quantities, including transition dynamics, rewards and policy values. This paper shows that such an approximation error perspective is *overly demanding* when attempting to characterize the conditions that lead to global convergence of PG methods.

Using the concept of approximation error, global convergence results for PG methods have been recently established in an additive form,

$$\text{sub-optimality gap} \leq \text{optimization error} + \text{approximation error},\tag{1}$$

implying that if the approximation error is small, a diminishing optimization error implies a small sub-optimality gap. A representative result is the global convergence of natural policy gradient (NPG) [4, Table 2], where the optimization error will diminish as the algorithm updates. There have also been global convergence results for other PG variants under linear function approximation that follow a similar approximation error analysis [3, 8, 10, 28, 5, 1, 2]. However, an additive bound like Eq. (1) has the inherent weakness that the approximation error will never be zero if the function approximation is not able to perfectly represent the desired quantities. This prevents such a strategy from establishing global convergence in cases where the approximation error is non-zero but a PG method still reaches the best representable solution.

Therefore, in spite of this recent progress, using approximation error in PG global convergence with function approximations has left two major gaps in the literature. First, it has not been investigated whether small approximation error is *necessary* to achieve convergence to an optimal representable policy [4], diverting attention from feature designs that achieve useful properties beyond small approximation error. Second, it is not clear if standard Softmax PG (other than NPG) converges globally under small approximation errors. In particular, NPG contains a least squares regression step [4, Eq. (17)] that can be naturally characterized with an approximation error quantity. However, standard Softmax PG does not have such a projection step [25], and the results in [4] do not apply to this update. Whether standard Softmax PG can achieve global convergence with even linearly realizable rewards (zero approximation error) is still an open problem.

In this paper, we address the above questions and contribute the following results. First, we provide negative answers to questions on the role of approximation error in determining global convergence of PG methods:

**(i)** Global convergence can be achieved under linear function approximation with non-zero approximation error, for both the standard Softmax PG and natural policy gradient (NPG) updates.
**(ii)** Approximation error is not a key quantity for characterizing global convergence in either case.
**(iii)** The conditions that imply global convergence are different between these two algorithms.

Second, these results lead us to question whether approximation error is an appropriate quantity to consider the global convergence of PG methods under linear function approximation. We establish new general results that characterize the conditions for global convergence of PG methods:

**(i)** NPG with log-linear function approximation achieves global convergence if and only if the projection of the reward onto the representable space preserves the optimal action's rank. This result significantly extends previous results that use approximation error in the analysis [4, 3], since preserving the rank of the optimal action is not strongly related to approximation error (except in the realizable limit).
**(ii)** We show that the global convergence of Softmax PG follows if the representation satisfies a non-domination condition and can preserve the ranking of rewards, which goes well beyond policy or reward realizability. As a byproduct, we resolve an open question by showing that even for linearly realizable reward function, Softmax PG cannot always converge to globally optimal policies when the non-domination condition for representation is violated.

## 2 Settings

We study the policy optimization problem under one state with $K$ actions. Given a reward vector $r \in \mathbb{R}^K$, the problem is to find a parametric policy $\pi_\theta$ to maximize the expected reward,

$$\sup_{\theta \in \mathbb{R}^d} \pi_\theta^\top r, \tag{2}$$

where $\theta \in \mathbb{R}^d$ with $d < K$ is the parameter, and $\pi_\theta = \text{softmax}(X\theta)$ is called a "log-linear policy" [4, 28] such that for all action $a \in [K] := \{1, 2, \ldots, K\}$,

$$\pi_\theta(a) = \frac{\exp\{[X\theta](a)\}}{\sum_{a' \in [K]} \exp\{[X\theta](a')\}}, \tag{3}$$

where $X \in \mathbb{R}^{K \times d}$ is the feature matrix with full column rank $d < K$. There are two major difficulties with the policy optimization problem. First, Eq. (2) is a non-concave maximization w.r.t. $\theta$, due to the softmax transform [22, Proposition 1]. Second, the policy and reward can be unrealizable, in the sense that the parametric log-linear policy $\pi_\theta = \text{softmax}(X\theta)$ cannot well approximate every policy $\pi$ in the $K$-dimensional probability simplex, and the score $X\theta \in \mathbb{R}^K$ cannot well approximate the true mean reward $r \in \mathbb{R}^K$. Such limitations arise in the linear function approximation case because $\pi_\theta$ and $X\theta$ are restricted to low-dimensional manifolds via $\theta \in \mathbb{R}^d$ for $d < K$.

To solve Eq. (2), we consider the standard Softmax PG [25] and NPG [13, 4] methods, shown in Algorithms 1 and 2. Softmax PG is an instance of gradient ascent, obtained by the chain rule,

$$\frac{d \pi_{\theta_t}^\top r}{d\theta_t} = \frac{d X\theta_t}{d\theta_t} \left( \frac{d \pi_{\theta_t}}{d X\theta_t} \right)^\top \frac{d \pi_{\theta_t}^\top r}{d\pi_{\theta_t}} = X^\top (\text{diag}(\pi_{\theta_t}) - \pi_{\theta_t} \pi_{\theta_t}^\top) r. \tag{4}$$

On the other hand, NPG conducts updates using least squares regression (i.e., projection),

$$\left( X^\top X \right)^{-1} X^\top r = \arg\min_{w \in \mathbb{R}^d} \|Xw - r\|_2^2. \tag{5}$$

As representative policy-based methods, in their general forms, Softmax PG and NPG lay the foundation for widely used RL methods, including REINFORCE [26], actor-critic [16, 7, 12], TRPO and PPO [23, 24]. The above Eqs. (4) and (5) are their updates applied to the one-state setting.

---

**Algorithm 1** Softmax policy gradient (PG)

**Input:** Learning rate $\eta > 0$.
**Output:** Policies $\pi_{\theta_t} = \text{softmax}(X\theta_t)$.
Initialize parameter $\theta_1 \in \mathbb{R}^d$.
**while** $t \geq 1$ **do**
$\quad \theta_{t+1} \leftarrow \theta_t + \eta \cdot X^\top (\text{diag}(\pi_{\theta_t}) - \pi_{\theta_t} \pi_{\theta_t}^\top) r$.
**end while**

---

**Algorithm 2** Natural policy gradient (NPG)

**Input:** Learning rate $\eta > 0$.
**Output:** Policies $\pi_{\theta_t} = \text{softmax}(X\theta_t)$.
Initialize parameter $\theta_1 \in \mathbb{R}^d$.
**while** $t \geq 1$ **do**
$\quad \theta_{t+1} \leftarrow \theta_t + \eta \cdot (X^\top X)^{-1} X^\top r$.
**end while**

---

To understand the difficulty of the optimization problem in Eq. (2), it is helpful to consider previous work that has analyzed the convergence of PG methods.

In the tabular setting, where $d = K$, $X = \text{Id}$, and $\pi_\theta = \text{softmax}(\theta)$ with $\theta \in \mathbb{R}^K$, both the rewards and optimal policy can be arbitrarily well approximated. In this case it is known that NPG enjoys a $O(1/t)$ global convergence rate [4, Table 1], which has been recently improved to $O(e^{-c \cdot t})$ [14, 20, 17, 27]. For the case of function approximation, such results have subsequently been extended to log-linear policies, where approximation error is used to characterize the projection step of Eq. (5) [4, 28]. In particular, NPG achieves the following sub-optimality gap for all $t \geq 1$ [4, Table 2],

$$(\pi^* - \pi_{\theta_t})^\top r \leq c_1/\sqrt{t} + c_2 \cdot \epsilon_{\text{approx}}, \qquad (c_1 > 0, \ c_2 > 0) \tag{6}$$

where $c_1$ and $c_2$ are problem specific constants, $\pi^*$ is the globally optimal policy, $\pi_{\theta_t}$ is produced by NPG, and $\epsilon_{\text{approx}}$ is the approximation error, i.e., the minimum error with which the policy values can be approximated using the features [4, Table 2]. The "optimization error" term $c_1/\sqrt{t}$ in Eq. (6) has since been improved to $O(e^{-c_3 \cdot t})$ with $c_3 > 0$ in [28, 5]. Note that if $\epsilon_{\text{approx}} > 0$ then Eq. (6) is insufficient for establishing $\pi_{\theta_t}^\top r \to r(a^*) := \max_{a \in [K]} r(a)$ as $t \to \infty$ even when such global convergence is achieved.

The understanding for the standard Softmax PG is even less clear. In the tabular case, it is known that Softmax PG achieves global convergence asymptotically, i.e., $\pi_{\theta_t}^\top r \to r(a^*)$ as $t \to \infty$ [4], with an $O(1/t)$ rate of convergence that exhibits undesirable problem and initialization dependent constants

[21, 18]. Directly extending this global convergence result to the case of function approximation, i.e., log-linear policies, is impossible without any additional assumptions on the features, since there can be exponentially many sub-optimal local maxima in the worst case [9]. In fact, even with linearly realizable rewards (zero approximation error), whether standard Softmax PG achieves global convergence still remains unsolved [4]. One intuitive reason why this is a difficult result to establish is that standard Softmax PG uses the gradient Eq. (4) rather than projection (regression) to perform updates, which is less directly connected to the concept of approximation error.

## 3 The Limitations of Approximation Error in Characterizing Convergence

It is known that there exist representations $X \in \mathbb{R}^{K \times d}$ with $d < K$ and $r \in \mathbb{R}^K$ that create exponentially many sub-optimal local maxima in Eq. (2) [9, Theorem 1], which makes it impossible to ensure global convergence of PG methods without imposing any structure on the function approximation. Before identifying specific conditions that ensure global convergence, we first explain how approximation error cannot be a useful structural measure for this purpose, by demonstrating that zero approximation error is not a necessary condition for global convergence, and illustrating problem instances with comparable approximation error that render starkly different convergence behaviors across different PG methods. Specifically, we illustrate these points with a set of concrete scenarios, each with 4 actions and 2-dimensional feature vectors describing each action. Since $d < K$, not every policy can be expressed in these representations, hence the problem instances are unrealizable.

### 3.1 Global Convergence is Achievable with Non-zero Approximation Error

The results of [9, Theorem 1] do not imply that sub-optimal local maxima always appear, as shown in the following.

**Example 1.** $K = 4$, $d = 2$, $X^\top = \begin{bmatrix} 0 & -1 & 0 & 2 \\ -2 & 0 & 1 & 0 \end{bmatrix}$ and $r = (9, 8, 7, 6)^\top$. The approximation error is $\epsilon_{approx} = \min\limits_{w \in \mathbb{R}^d} \|Xw - r\|_2 = \left\| X \left( X^\top X \right)^{-1} X^\top r - r \right\|_2 = \sqrt{202.6} \approx 14.2338$.

Note that the approximation error is larger than any sub-optimality gap, i.e., for any policy $\pi$,

$$(\pi^* - \pi)^\top r \leq 3 < \epsilon_{\text{approx}}, \tag{7}$$

hence the bound in Eq. (6) does not imply global convergence for NPG in this example. Yet, despite the non-zero approximation error and the inability of existing results including Eq. (6) to establish global convergence on Example 1, both Algorithms 1 and 2 can be shown to reach a global maximum.

**Proposition 1.** Denote $a^* := \arg\max_{a \in [K]} r(a)$. With constant $\eta > 0$ and any initialization $\theta_1 \in \mathbb{R}^d$, both Algorithms 1 and 2 guarantee $\pi_{\theta_t}^\top r \to r(a^*)$ as $t \to \infty$ on Example 1.

All proofs can be found in the appendix due to space limits. The fact that Softmax PG achieves global convergence in Example 1 is much harder to establish than for NPG, since Eq. (4) involves a complex non-linearity given the presence of the softmax, unlike the linear least squares Eq. (5) used in NPG. To illustrate the intuition behind Proposition 1 we use a visualization of the optimization landscape.

**Visualization.** A visualization of the optimization landscape of Example 1 is shown in Figure 1(a). The bottom two-dimensional plane is the parameter space $\mathbb{R}^d$ where $d = 2$. For each $\theta \in \mathbb{R}^d$, we calculate $\pi_\theta$ using Eq. (3) and $\pi_\theta^\top r$ using Eq. (2), and use $\pi_\theta^\top r$ as the vertical axis value of $\theta$.

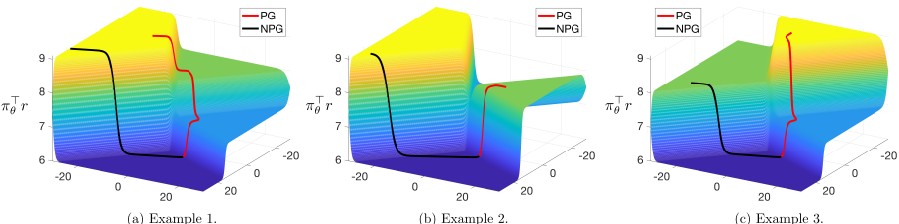

Figure 1: Visualizing the landscapes in the example problem instances.

To verify Proposition 1, we run Softmax PG and NPG on Example 1 with the same $\theta_1 = (6, 8)^\top \in \mathbb{R}^2$. In Figure 1(a), the optimization trajectories show 85 iterations of NPG and $8.5 \times 10^6$ iterations of

Softmax PG, both with learning rate $\eta = 0.2$. It can be clearly seen that both Softmax PG and NPG eventually achieve expected reward $\pi_{\theta_t}^\top r \to 9 = r(a^*)$, demonstrating global convergence (Figure 3(c) later shows that the sub-optimality gap $(\pi^* - \pi_{\theta_t})^\top r$ approaches 0).

In summary, Example 1 shows that both Softmax PG and NPG are able to achieve global convergence on unrealizable problem instances with non-zero approximation error. This raises the question:

*Is non-zero approximation error useful for characterizing global convergence?*

## 3.2 Global Convergence is Irrelevant to Non-zero Approximation Error

We answer the above question negatively. By comparing alternative problem instances with similar approximation errors but different convergence behaviors, we illustrate how approximation error is not able to distinguish between scenarios where global versus local convergence is obtained.

**Example 2.** $K = 4$, $d = 2$, $X^\top = \begin{bmatrix} 0 & 0 & -1 & 2 \\ -2 & 1 & 0 & 0 \end{bmatrix} \in \mathbb{R}^{d \times K}$, and $r = (9, 8, 7, 6)^\top \in \mathbb{R}^K$. The approximation error is $\left\| X \left( X^\top X \right)^{-1} X^\top r - r \right\|_2 = \sqrt{205} \approx 14.3178$.

The only difference between Examples 1 and 2 is that the second and third columns of $X^\top$ have been exchanged. The approximation error remains similar to that of Example 1. Using the upper bound of Eq. (6), one might therefore expect similar sub-optimality gaps $(\pi^* - \pi_{\theta_t})^\top r$ to be demonstrated by the algorithms, since the r.h.s. contains similar approximation errors. However, as shown in Figure 1(b), using the same initialization and learning rate, Softmax PG obtains $\pi_{\theta_t}^\top r \to 8 = r(2) < r(a^*)$ as it converges to a sub-optimal deterministic policy, while NPG continues to succeed.

Lest one believe that NPG is globally convergent, the following example, where the first and second columns of $X^\top$ are swapped, illustrate an analogous failure for NPG but not Softmax PG.

**Example 3.** $K = 4$, $d = 2$, $X^\top = \begin{bmatrix} -1 & 0 & 0 & 2 \\ 0 & -2 & 1 & 0 \end{bmatrix} \in \mathbb{R}^{d \times K}$, and $r = (9, 8, 7, 6)^\top \in \mathbb{R}^K$. The approximation error is $\left\| X \left( X^\top X \right)^{-1} X^\top r - r \right\|_2 = \sqrt{212} \approx 14.5602$.

Here again the approximation error is close to that of Example 1. Yet, Figure 1(c) shows that NPG achieves $\pi_{\theta_t}^\top r \to 8 < r(a^*)$ as it converges to a sub-optimal solution, while Softmax PG succeeds.

In summary, the Examples 1, 2 and 3 all have similar approximation errors, yet Softmax PG achieves global convergence on Example 1 but reaches a bad local maxima on Example 2, while NPG succeeds on Example 1 and fails on Example 3. Note that these examples can be re-scaled to have exactly the same approximation errors while demonstrating the same convergence behavior of the algorithms. From these findings we conclude that, if there is any quantity that can predict whether global versus local convergence is obtained by Softmax PG or NPG, that the quantity cannot be approximation error alone. This motivates to investigate the question: what is the right quantity to characterize global convergence for unrealizable problems?

## 3.3 Global Convergence Characterization is Algorithm Dependent

We make one more key point. From Figure 1(b) and Figure 1(c), NPG achieves global convergence on Example 2 but fails on Example 3, while, conversely, Softmax PG succeeds on Example 3 and fails on Example 2. This difference indicates that whatever condition characterizes global convergence, it must be *algorithm dependent*, even for the closely related algorithms Softmax PG and NPG. Therefore, one has to study the conditions for Softmax PG and NPG **respectively** (rather than one condition for both algorithms), which motivates the refined question:

*What conditions characterize global convergence of Softmax PG and NPG in unrealizable problems?*

# 4 New Characterizations of Global Convergence for PG Methods

From these observations, it is clear that whatever quantity characterizes the global convergence of PG methods, it cannot be based solely on approximation error and it must be algorithm dependent. Therefore, we study distinct global convergence conditions for Softmax PG and NPG respectively.

## 4.1 Reward Order Preservation with Adequate Features is Sufficient for PG Convergence

We now investigate a global convergence condition for Softmax PG under log-linear policies.

**Intuition.** Consider Example 1, where Softmax PG achieves global convergence. From the landscape shown in Figure 1(a), there appears to be a monotonic path from any initialization point that allows gradient ascent to reach the optimal plateau with reward $r(a^*) = 9$. Intuitively, this arises because the actions' rewards seem to be nicely "ordered". For example, starting from $\theta_1 = (6, 8)^\top \in \mathbb{R}^d$ such that $\pi_{\theta_1}^\top r \approx 6$, Softmax PG is able to improve its expected reward eventually to $\pi_{\theta_t}^\top r \approx 7$, since there exists a sub-optimal plateau with a higher reward 7 right beside the lowest plateau with reward 6. Next, Softmax PG continues to improve its expected reward eventually to $\pi_{\theta_t}^\top r \approx 8$ by "climbing" toward another neighboring plateau with a higher reward. Finally, this process ends with Softmax PG successfully arriving at the optimal plateau with reward $r(a^*) = 9$.

By contrast, in Example 2, as shown in Figure 1(b), Softmax PG gets stuck on a bad plateau with a local maximum reward of 8. Visually, Softmax PG stops improving its expected reward on this sub-optimal plateau, because it is "surrounded" by two lower plateaus with rewards 6 and 7, which breaks the nice "ordering" of the expected reward landscape and traps the gradient ascent trajectory on a sub-optimal plateau from which there is no monotonic ascent to global optimality.

**Verifying reward order preservation.** Based on the above intuition and observations, we conjecture that the ordering structure between the different rewards is a key property behind the global convergence of Softmax PG. We can verify this conjecture in each of the Examples 1 to 3 by determining whether the feature matrix $X \in \mathbb{R}^{K \times d}$ allows the same action ordering as the reward vector $r \in \mathbb{R}^K$ to be realized. For Example 1, note that with $w = (-1, -1)^\top \in \mathbb{R}^d$, we have

$$r' := Xw = (2, 1, -1, -2)^\top \in \mathbb{R}^K, \tag{8}$$

which preserves the ordering of $r \in \mathbb{R}^K$, such that for all $i, j \in [K]$, $r(i) > r(j)$ if and only if $r'(i) > r'(j)$. Similarly, for Example 3, if we let $w = (-3, -1)^\top$ then we have $r' := Xw = (3, 2, -1, -6)^\top$, which also preserves the order of $r$ over actions. Softmax PG converges to a globally optimal reward in both of these examples.

By contrast, for Example 2, it is impossible to find any $w \in \mathbb{R}^d$ such that $Xw$ preserves the order of the rewards $r$. To see why, consider any $w = (w(1), w(2))^\top$ and note that

$$r' := Xw = (-2 \cdot w(2), w(2), -w(1), 2 \cdot w(1))^\top. \tag{9}$$

To preserve the reward order, we require both $-2 \cdot w(2) > w(2)$ (which would imply $w(2) < 0$) and $-w(1) > 2 \cdot w(1)$ (which would imply $w(1) < 0$), but these two conditions imply $w(2) < 0 < -w(1)$, which must reverse the order of the second and third actions. This is an example where PG can fail to reach a global optimum.

**Main Softmax PG result.** We formalize the above intuition by proving the following main result, which establishes that reward order preservation with adequate representations is a sufficient condition for the global convergence of Softmax PG under log-linear function approximation.

**Theorem 1** (Reward order preservation, non-domination features)**.** *Given any reward $r \in \mathbb{R}^K$ and feature matrix $X \in \mathbb{R}^{K \times d}$. Denote $x_i \in \mathbb{R}^d$ as the $i$-th row vector of $X$. If (i) $x_i^\top x_i > x_i^\top x_j$ for all $j \neq i$, and (ii) there exists at least one $w \in \mathbb{R}^d$, s.t., $r' := Xw$ preserves the order of $r$, i.e., for all $i, j \in [K]$, $r(i) > r(j)$ if and only if $r'(i) > r'(j)$, then for any initialization $\theta_1 \in \mathbb{R}^d$, Algorithm 1 with a constant learning rate $\eta > 0$ achieves global convergence of $\pi_{\theta_t}^\top r \to r(a^*)$ as $t \to \infty$.*

A few remarks about this theorem are in order.

Examples 1 to 3 all satisfy the non-domination condition **(i)** on $X$, and their differences lie in satisfying reward order preservation or not. However, the following example shows that if the condition **(i)** on $X$ is removed, then global convergence is not always achievable for even linearly realizable rewards (with zero approximation error).

**Proposition 2.** *Let $K = 3$, $d = 2$, $X^\top = \begin{bmatrix} 0 & -10 & 0 \\ -2 & 4 & 1 \end{bmatrix} \in \mathbb{R}^{d \times K}$, and $r = Xw = (4, 2, -2)^\top$, where $w = (-1, -2)^\top \in \mathbb{R}^d$. With initialization $\theta_1 = (-\ln 2, \ln 2)^\top$, Algorithm 1 does not achieve global convergence, i.e., $\pi_{\theta_t}(1) \not\to 1$ as $t \to \infty$.*

**Generalization of tabular and linear realizability.** When $d = K$ and $X = \mathbf{Id}$, i.e., the softmax tabular parameterization $\pi_\theta = \text{softmax}(\theta)$, it is always true that $Xr = r$ preserves the order of $r$. Consequently, Theorem 1 recovers the global convergence result for PG in the softmax tabular setting [4, 22] as a special case. More generally, for non-domination features, when the reward is linearly realizable, such that $Xw = r$ for some $w \in \mathbb{R}^d$, the global convergence of Softmax PG also follows from Theorem 1, since $r$ preserves its own order when the approximation error is zero.

**Corollary 1** (Linearly realizable rewards, non-domination features). *Given any reward $r \in \mathbb{R}^K$ and feature matrix $X \in \mathbb{R}^{K \times d}$. Denote $x_i \in \mathbb{R}^d$ as i-th row vector of X. If **(i)** $x_i^\top x_i > x_i^\top x_j$ for all $j \neq i$, and **(ii)** there exists $w \in \mathbb{R}^d$, s.t., $Xw = r$, then for any initialization $\theta_1 \in \mathbb{R}^d$, Algorithm 1 with a constant learning rate $\eta > 0$ achieves global convergence of $\pi_{\theta_t}^\top r \to r(a^*)$ as $t \to \infty$.*

It is worth mentioning that Proposition 2 and Corollary 1 together answer a question which still remain unsolved in PG literature [4]: with linearly realizable rewards (zero approximation error), whether standard Softmax PG achieves global convergence? Proposition 2 shows that linearly realizable reward on its own is not enough to guarantee global convergence, while Corollary 1 shows that with adequate features, linearly realizable reward implies global convergence. Note that the NPG global convergence result in [4], such as Eq. (6), does not apply to standard Softmax PG.

**Ordering does not determine approximation.** As already illustrated in Section 3, approximation error is not adequate for capturing the global convergence of Softmax PG. It is important to emphasize that the existence of an order preserving reward $r'$ is very different from having a small approximation error. When the approximation error is zero, then an order preserving reward (equal to $r$) always exists. However, in general, $r'$ can take very different values than $r$, and hence have a very large approximation error, yet still enable global convergence as shown in Examples 1 and 3.

**Proof idea.** The idea behind the proof of the main theorem consists of three parts. We provide a sketch of the proof here; the full proof is given in Appendix A. **First**, starting from any initialization $\theta_t \in \mathbb{R}^d$, Algorithm 1 guarantees that $\pi_{\theta_t}$ will approach a (generalized) one-hot policy as $t \to \infty$. To see why, first note that $\pi_\theta^\top r$ is $\beta$-smooth over $\theta \in \mathbb{R}^d$ with some $\beta > 0$ (Lemma 3 in Appendix B), since the softmax transform is smooth [4, 22] and the feature matrix $X$ has bounded values. This implies that using a sufficiently small constant learning rate $0 < \eta \leq 2/\beta$ we obtain,

$$\pi_{\theta_{t+1}}^\top r - \pi_{\theta_t}^\top r \geq \frac{1}{2\beta} \cdot \left\| \frac{d\,\pi_{\theta_t}^\top r}{d\theta_t} \right\|_2^2 \geq 0. \tag{10}$$

Note that $\pi_\theta^\top r$ is upper bounded by $r(a^*)$. According to the monotone convergence, $\pi_{\theta_t}^\top r \to c \leq r(a^*)$ as $t \to \infty$. This fact combined with Eq. (10) implies $\left\| \frac{d\,\pi_{\theta_t}^\top r}{d\theta_t} \right\|_2 \to 0$ as $t \to \infty$. Next, a special co-variance structure of softmax PG (Lemma 4) shows that $\left\| \frac{d\,\pi_{\theta_t}^\top r}{d\theta_t} \right\|_2 \to 0$ implies that $\|\theta_t\|_2 \to \infty$ and $\pi_{\theta_t}$ approaches a (generalized) one-hot policy as $t \to \infty$.

**Lemma 1.** *Under the same conditions as Theorem 1, and $r(i) \neq r(j)$ for all $i \neq j$ (unique action reward), Algorithm 1 assures $\|\theta_t\|_2 \to \infty$ and $\pi_{\theta_t}(i) \to 1$ for an action $i \in [K]$ as $t \to \infty$.*

**Remark 1.** *Removing the unique action reward condition in Lemma 1 makes $\pi_{\theta_t}$ approach a generalized one-hot policy (rather than a strict one-hot in Lemma 1) as $t \to \infty$ as a result.*

According to Lemma 1, $\theta_t$ grows unboundedly. Intuitively, this can be seen in Figure 1(a), where there are no stationary points in a finite region.

**Second**, for any vector $r'$ that preserves the order of $r$, we establish the following key lemma.

**Lemma 2** (Non-negative covariance of order preservation). *If $r' \in \mathbb{R}^K$ preserves the order of $r \in \mathbb{R}^K$, i.e., for all $i, j \in [K]$, $r(i) > r(j)$ iff $r'(i) > r'(j)$, then for any policy $\pi \in \Delta(K)$,*

$$r'^\top \left( diag(\pi) - \pi\pi^\top \right) r = \text{Cov}_\pi \left( r', r \right) \geq 0. \tag{11}$$

Now consider the direction $w \in \mathbb{R}^d$ such that $r' := Xw$ preserves the order of $r$. We have,

$$w^\top \theta_{t+1} = w^\top \theta_t + \eta \cdot w^\top X^\top \left( \text{diag}(\pi_{\theta_t}) - \pi_{\theta_t}\pi_{\theta_t}^\top \right) r \qquad \text{(by Algorithm 1)} \tag{12}$$

$$= w^\top \theta_t + \eta \cdot r'^\top \left( \text{diag}(\pi_{\theta_t}) - \pi_{\theta_t}\pi_{\theta_t}^\top \right) r \qquad (r' := Xw) \tag{13}$$

$$\geq w^\top \theta_t. \qquad \text{(by Lemma 2)} \tag{14}$$

**Third**, take a sub-optimal action $i \in [K]$ with $r(i) < r(a^*)$, and we show that the assumption $\pi_{\theta_t}(i) \to 1$ as $t \to \infty$ leads to a contradiction.

To that end, first observe that this assumption implies that for all large enough time $t \geq 1$,

$$\left[\frac{X\theta_t}{\|\theta_t\|_2}\right](i) = \max_{a \in [K]} \left[\frac{X\theta_t}{\|\theta_t\|_2}\right](a), \tag{15}$$

which means that the sub-optimal action $i \in [K]$ always has the largest score (since its probability $\pi_{\theta_t}(i) \to 1$ is always the largest). Moreover, differences between actions' scores are unbounded, due to $\frac{\pi_{\theta_t}(i)}{\pi_{\theta_t}(j)} = \exp\{[X\theta_t](i) - [X\theta_t](j)\} \to \infty$ for all other actions $j \neq i$.

Consider Example 1 for illustration. The top view of Figure 1(a) is shown in Figure 2(a). Take $i = 2$ and $r(i) = 8$, and assume $\frac{\theta_t}{\|\theta_t\|_2}$ stays in the green (sub-optimal) region of Figure 2(a), excluding its boundaries. This green region is partitioned in Figure 2(b), where the dark sub-region contains $v_2 \in \mathbb{R}^d$ such that $[Xv_2](a^*)$ is the second largest component among all $a \neq 2$, and the light sub-region is the remaining.

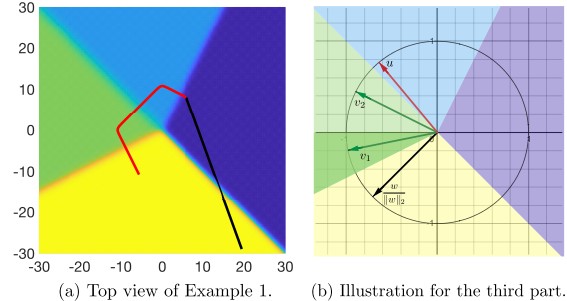

(a) Top view of Example 1.    (b) Illustration for the third part.

Figure 2: Idea illustration.

The argument is completed by addressing the two cases: **(i)** If $\frac{\theta_t}{\|\theta_t\|_2}$ stays in the dark sub-region where $v_1 \in \mathbb{R}^d$ belongs to, then $\pi_{\theta_t}^\top r > r(i) = 8$ must occur in finite $t < \infty$, implying $\pi_{\theta_t}(i) \not\to 1$, contradicting the assumption. Intuitively, the contradiction occurs because the dark sub-region is closer to a higher plateau with reward 9, and scaling up $\theta_t$'s magnitude in this sub-region eventually ensures $\pi_{\theta_t}^\top r > r(i) = 8$. **(ii)** If $\frac{\theta_t}{\|\theta_t\|_2}$ stays in the light sub-region which contains $v_2 \in \mathbb{R}^d$, then $w^\top \theta_t > u^\top \theta_t$ must occur in finite time $t < \infty$, implying that $\frac{\theta_t}{\|\theta_t\|_2}$ will enter the dark sub-region, reducing to the first case. This argument depends on Eq. (12) and a key observation showing that $u^\top \theta_{t+1} < u^\top \theta_t$, where $u$ is a "worse" direction such that $[Xu](a^-) = \max_{a \in [K]}[Xu](a)$ for some $a^- \in [K]$ with $r(a^-) < r(i)$.

To summarize, $\frac{\theta_t}{\|\theta_t\|_2}$ cannot always stay in the green sub-optimal region in Figure 2(a), which implies that $\frac{\theta_t}{\|\theta_t\|_2}$ must eventually enter the optimal region that contains $w$ and stay in that region. By Lemma 1 we then obtain $\pi_{\theta_t}(a^*) \to 1$ and $\pi_{\theta_t}^\top r \to r(a^*)$ as $t \to \infty$ (see appendix).

## 4.2 Optimal Action Preservation is Necessary and Sufficient for NPG Convergence

Next, we investigate the global convergence conditions for NPG under log-linear policies. Unlike Softmax PG, the key property for determining global convergence of NPG is whether the projection of the rewards $r$ onto the feature representation $X$ preserves the top ranking of the optimal action.

**Intuition and demonstration.** First consider Example 1 where NPG successfully converges to a global maximum. From Algorithm 2, a simple calculation shows,

$$X\theta_{t+1} = X\theta_t + \eta \cdot X(X^\top X)^{-1}X^\top r = X\theta_t + \eta \cdot \frac{1}{5} \cdot (22, -4, -11, 8)^\top, \tag{16}$$

which implies that the optimal action $a^* = 1$ always receives the largest update to its score $[X\theta_t](a^*)$ in each iteration. Next, take a sub-optimal action $a = 2$, as an example, and observe that,

$$\frac{\pi_{\theta_{t+1}}(a^*)}{\pi_{\theta_{t+1}}(a)} = \frac{\pi_{\theta_t}(a^*)}{\pi_{\theta_t}(a)} \cdot \exp\{\eta \cdot (\hat{r}(a^*) - \hat{r}(a))\} = \frac{\pi_{\theta_t}(a^*)}{\pi_{\theta_t}(a)} \cdot \exp\left\{\eta \cdot \frac{26}{5}\right\} \tag{17}$$

by Eq. (3), where $\hat{r} := X(X^\top X)^{-1}X^\top r$. Using a constant learning rate $\eta > 0$ and applying Eq. (17), we have that $\pi_{\theta_t}(a^*)$ grows exponentially with $t$, indicating that $\pi_{\theta_t}(a^*) \to 1$ as $t \to \infty$ since the same argument works for any sup-optimal action $a \neq a^*$. Moreover, the rate is $O(e^{-c \cdot t})$, since $(\pi^* - \pi_{\theta_t})^\top r \leq 2 \cdot \|r\|_\infty \cdot (1 - \pi_{\theta_t}(a^*))$. The $O(e^{-c \cdot t})$ rate matches the results in softmax tabular settings [14, 20, 17, 27].

Second, consider Example 2 where NPG fails to converge to a global maximum. Using similar calculations to Eq. (16) we obtain,

$$X\theta_{t+1} = X\theta_t + \eta \cdot \hat{r} = X\theta_t + \eta \cdot \frac{1}{5} \cdot (-3, 18, -9, -6)^\top \tag{18}$$

which implies that a sub-optimal action $a = 2$ always receives the largest update on its score $[X\theta_t](2)$ in each iteration. The failure in Figure 1(b) is then verified by similar arguments around Eq. (17).

**Main NPG result.** Based on these observations, it is evident that for NPG to converge globally, it is important for the optimal action to eventually always receive the largest update to its score, which makes it critical that the least square projection $X(X^\top X)^{-1}X^\top r$ preserves the top ranking of the optimal action. We formalize this intuition by establishing the following main result.

**Theorem 2** (Optimal action preservation condition). *For a constant learning rate $\eta > 0$, a necessary and sufficient condition for Algorithm 2 to achieve global convergence $\pi_{\theta_t}^\top r \to r(a^*)$ as $t \to \infty$ from any initialization $\theta_1 \in \mathbb{R}^d$ is that $\hat{r}(a^*) > \hat{r}(a)$ for all $a \neq a^*$, such that $a^* := \arg\max_{a \in [K]} r(a)$, and $\hat{r} := X(X^\top X)^{-1}X^\top r$ is the least squares projection of $r$ onto the column space of $X$. If the condition is satisfied, then the rate of convergence is $(\pi^* - \pi_{\theta_t})^\top r \in O(e^{-c \cdot t})$ for some $c > 0$.*

**Proof idea.** When the optimal action preservation is satisfied, similar arguments to Eqs. (16) and (17) guarantee that $\pi_{\theta_t}(a^*)$ grows exponentially with $t$, indicating that $\pi_{\theta_t}(a^*) \to 1$ as $t \to \infty$.

The constant $c > 0$ in Theorem 2 depends on the gap of $\hat{r}$, i.e., $\hat{r}(a^*) - \max_{a \neq a^*} \hat{r}(a)$, which finds similarities to NPG results in tabular settings [14, 15]. The main difference is that the gap of true reward $r$ in tabular cases is replaced with the gap of least square projection $\hat{r}$ in function approximation settings in Theorem 2. This similarity is an evidence for improving the rate to super-linear by using geometrically increasing step sizes, as in tabular settings [17, 27, 19, 28, 6].

**One-sided Approximation Error.** For NPG, [4, Lemma 6.2] introduces a "one-sided approximation error" quantity, which aims to overestimate the advantage of the optimal action $a^*$,

$$\epsilon_t := r(a^*) - \pi_{\theta_t}^\top r - w^\top (x_{a^*} - X^\top \pi_{\theta_t}) = r(a^*) - \pi_{\theta_t}^\top r - (\hat{r}(a^*) - \pi_{\theta_t}^\top \hat{r}). \tag{19}$$

This quantity relaxes the notion of approximation error and still guarantees the global convergence of NPG, since if $\sum_{t=1} \epsilon_t \in o(T)$, then NPG with $\eta \in O(1/\sqrt{T})$ achieves global convergence [4, Lemma 6.2]. We note however that Eq. (19) has two limitations: **(i)** Eq. (19) depends on the entire update trajectory $\{\theta_t\}_{t \geq 1}$, which is hard to verify. By contrast, the optimal action preservation condition in Theorem 2 only involves problem quantities $X$ and $r$. **(ii)** It is not clear whether Eq. (19) is a necessary condition for global convergence, while optimal action preservation is proved above to be both necessary and sufficient.

## 5   Simulation Study

We conducted additional simulations to check the theoretical results. **First**, we check whether the strict inequality of $\hat{r}(a^*) > \hat{r}(a)$ for all $a \neq a^*$ in Theorem 2 is required for NPG global convergence.

**Example 4.** $K = 4$, $d = 2$, $X^\top = \begin{bmatrix} 0 & -1 & 0 & 1 \\ -1 & 0 & 1 & 0 \end{bmatrix} \in \mathbb{R}^{d \times K}$, and $r = (9, 8, 7, 6)^\top \in \mathbb{R}^K$. The best fit for $r$ is $\hat{r} = X(X^\top X)^{-1}X^\top r = (1, 1, -1, -1)^\top$.

Example 4 has $\hat{r}(a^*) = \hat{r}(1) = \hat{r}(2)$, which violates the strict inequality condition of $\hat{r}(a^*) > \hat{r}(a)$ for all $a \neq a^*$. The consequence is that NPG guarantees $\frac{\pi_{\theta_t}(a^*)}{\pi_{\theta_t}(2)} = \frac{\pi_{\theta_1}(a^*)}{\pi_{\theta_1}(2)}$ for all $t \geq 1$, which makes it impossible for $\pi_{\theta_t}(a^*) \to 1$ as $t \to \infty$. This is observed in Figure 3(a), supporting that the strictly inequality condition in Theorem 2 is indeed necessary. The initialization is $\theta_1 = (4, 10)^\top$, and $\eta = 0.2$. We run 150 iterations for NPG and $1.5 \times 10^7$ iterations for Softmax PG.

**Second**, we run 150 iterations of NPG on Example 1. As shown in Figure 3(b), the quantity $\log(\pi^* - \pi_{\theta_t})^\top r$ is a linear function of time $t$, implying that $(\pi^* - \pi_{\theta_t})^\top r \in O(e^{-c \cdot t})$ with $c > 0$. This supports the convergence rate results in Theorem 2. Here $\theta_1$ and $\eta$ are the same as in Figure 1(a).

**Third**, we check whether the condition in Theorem 1 is required for Softmax PG global convergence.

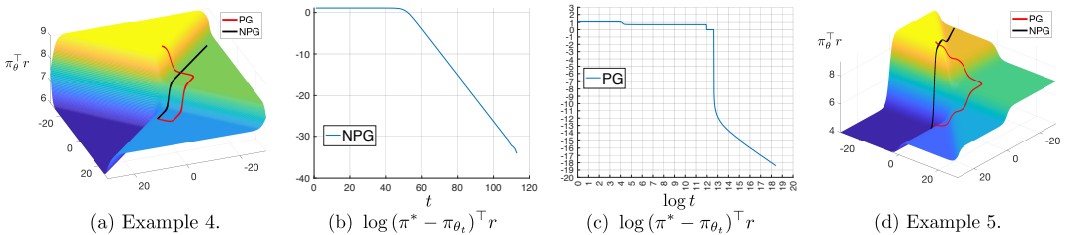

|       |       |       |       |
| ----- | ----- | ----- | ----- |
| (a) Example 4. | (b) $\log{(\pi^* - \pi_{\theta_t})}^\top r$ | (c) $\log{(\pi^* - \pi_{\theta_t})}^\top r$ | (d) Example 5. |

Figure 3: Simulations for verifying theoretical results.

**Example 5.** $K = 6$, $d = 2$, $X^\top = \begin{bmatrix} 0 & -1 & -1 & 0 & 1 & 1 \\ -1 & 0 & 1 & 1 & 0 & -1 \end{bmatrix}$, *and* $r = (9, 8, 7, 6, 5, 4)^\top$.

Similar to Eq. (9), it is impossible to find any $w \in \mathbb{R}^d$, such that $r' := Xw$ preserves the order of $r$ in Example 5. However, as shown in Figure 3(d), Softmax PG achieves $\pi_{\theta_t}^\top r \to r(a^*) = 9$, indicating that the reward order preservation condition in Theorem 1 is sufficient but not necessary for PG to achieve global convergence. The initialization is $\theta_1 = (10, -2)^\top$, and $\eta = 0.2$. We run 100 iterations for NPG and $2 \times 10^6$ iterations for Softmax PG. Note that NPG behaves erratically on Example 5 (which does not satisfy its global convergence conditions), by first entering then leaving the optimal plateau, eventually approaching a sub-optimal solution.

**Finally**, we run $10^8$ iterations of Softmax PG on Example 1, using the same $\eta$ and $\theta_1$ as in Figure 1(a). Figure 3(c) shows that the slope of $\log{(\pi^* - \pi_{\theta_t})}^\top r$ over $\log t$ approaches $-1$, indicating that the global convergence rate is $(\pi^* - \pi_{\theta_t})^\top r \in O(1/t)$, matching the softmax tabular setting results [22].

## 6 Discussions

**Checking ordering-based conditions.** Checking the existence of $w \in \mathbb{R}^d$ in Theorem 1 is known as linear feasibility in literature [11], i.e., determining whether a set of inequalities has a non-empty intersection. In particular, suppose $X \in \mathbb{R}^{K \times d}$, and $r \in \mathbb{R}^K$ is sorted, i.e., $r(1) \geq r(2) \geq \cdots \geq r(K)$. Denote $x_i \in \mathbb{R}^d$ as the $i$-th row vector of $X$. The linear feasibility problem in this case is to check if there exists $w \in \mathbb{R}^d$, such that for all $i \in [K-1]$, $x_i^\top w \geq x_{i+1}^\top w$. Linear feasibility can be cast as linear programming (LP) using a dummy objective and keeping the constraints, hence any LP technique, such as the ellipsoid method, can be used to solve it [11]. On the other hand, checking the optimal action preservation in Theorem 2 requires the same information as in calculating approximation error $\|\hat{r} - r\|_2 = \min_{w \in \mathbb{R}^d} \|Xw - r\|_2$, since $\arg\max_{a \in [K]} \hat{r}(a) = \arg\max_{a \in [K]} r(a)$ can be immediately verified after calculating the projection $\hat{r} := X^\top (X^\top X)^{-1} X^\top r$.

**Generalization to Markov decision processes (MDPs).** Our work provides some new and useful insights for understanding more complex settings, but it requires further investigation to resolve this highly non-trivial problem for general MDPs. See Appendix C for detailed discussions.

## 7 Conclusions and Future Work

We believe this work opens new directions for understanding PG-based methods under function approximation, going well beyond the conventional approximation error based analysis. The major technical findings involve ordering-based conditions and relevant techniques (covariance and global convergence). Identifying exact necessary and sufficient conditions for the global convergence of Softmax PG remains future work. Extending the results and techniques to general MDPs is another important and challenging next step. Combining function approximation with recent results on stochastic on-policy sampling [20] is another interesting direction for agnostic learning. Investigating whether these new global convergence conditions might be used to achieve better representation learning is of great interest for algorithm design. Generalizing the proof techniques to other scenarios where non-linear transforms (activation functions) interact with low-dimensional features through gradient descent, such as in neural networks, is another lofty ambition.

## Acknowledgments and Disclosure of Funding

The authors would like to thank anonymous reviewers for their valuable comments. CS, DS gratefully acknowledge funding from the Canada CIFAR AI Chairs Program, Amii and NSERC.

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
