\left(\pi^* - \pi_{\theta_t}\right)^\top r$ | (c) $\log \left(\pi^* - \pi_{\theta_t}\right)^\top r$ | (d) Example 5. |

Figure 3: Simulations for verifying theoretical results.

**Example 5.** $K = 6$, $d = 2$, $X^\top = \begin{bmatrix} 0 & -1 & -1 & 0 & 1 & 1 \\ -1 & 0 & 1 & 1 & 0 & -1 \end{bmatrix}$, *and* $r = (9, 8, 7, 6, 5, 4)^\top$.

Similar to Eq. (9), it is impossible to find any $w \in \mathbb{R}^d$, such that $r' := Xw$ preserves the order of $r$ in Example 5. However, as shown in Figure 3(d), Softmax PG achieves $\pi_{\theta_t}^\top \to r(a^*) = 9$, indicating that the reward order preservation condition in Theorem 1 is sufficient but not necessary for PG to achieve global convergence. The initialization is $\theta_1 = (10, -2)^\top$, and $\eta = 0.2$. We run 100 iterations for NPG and $2 \times 10^6$ iterations for Softmax PG. Note that NPG behaves erratically on Example 5 (which does not satisfy its global convergence conditions), by first entering then leaving the optimal plateau, eventually approaching a sub-optimal solution.

**Finally**, we run $10^8$ iterations of Softmax PG on Example 1, using the same $\eta$ and $\theta_1$ as in Figure 1(a). Figure 3(c) shows that the slope of $\log \left(\pi^* - \pi_{\theta_t}\right)^\top r$ over $\log t$ approaches $-1$, indicating that the global convergence rate is $\left(\pi^* - \pi_{\theta_t}\right)^\top r \in O(1/t)$, matching the softmax tabular setting results [22].

## 6 Discussions

**Checking ordering-based conditions.** Checking the existence of $w \in \mathbb{R}^d$ in Theorem 1 is known as linear feasibility in literature [11], i.e., determining whether a set of inequalities has a non-empty intersection. In particular, suppose $X \in \mathbb{R}^{K \times d}$, and $r \in \mathbb{R}^K$ is sorted, i.e., $r(1) \geq r(2) \geq \cdots \geq r(K)$. Denote $x_i \in \mathbb{R}^d$ as the $i$-th row vector of $X$. The linear feasibility problem in this case is to check if there exists $w \in \mathbb{R}^d$, such that for all $i \in [K-1]$, $x_i^\top w \geq x_{i+1}^\top w$. Linear feasibility can be cast as linear programming (LP) using a dummy objective and keeping the constraints, hence any LP technique, such as the ellipsoid method, can be used to solve it [11]. On the other hand, checking the optimal action preservation in Theorem 2 requires the same information as in calculating approximation error $\|\hat{r} - r\|_2 = \min_{w \in \mathbb{R}^d} \|Xw - r\|_2$, since $\arg\max_{a \in [K]} \hat{r}(a) = \arg\max_{a \in [K]} r(a)$ can be immediately verified after calculating the projection $\hat{r} := X^\top (X^\top X)^{-1} X^\top r$.

**Generalization to Markov decision processes (MDPs).** Our work provides some new and useful insights for understanding more complex settings, but it requires further investigation to resolve this highly non-trivial problem for general MDPs. See Appendix C for detailed discussions.

## 7 Conclusions and Future Work

We believe this work opens new directions for understanding PG-based methods under function approximation, going well beyond the conventional approximation error based analysis. The major technical findings involve ordering-based conditions and relevant techniques (covariance and global convergence). Identifying exact necessary and sufficient conditions for the global convergence of Softmax PG remains future work. Extending the results and techniques to general MDPs is another important and challenging next step. Combining function approximation with recent results on stochastic on-policy sampling [20] is another interesting direction for agnostic learning. Investigating whether these new global convergence conditions might be used to achieve better representation learning is of great interest for algorithm design. Generalizing the proof techniques to other scenarios where non-linear transforms (activation functions) interact with low-dimensional features through gradient descent, such as in neural networks, is another lofty ambition.

## Acknowledgments and Disclosure of Funding

The authors would like to thank anonymous reviewers for their valuable comments. CS, DS gratefully acknowledge funding from the Canada CIFAR AI Chairs Program, Amii and NSERC.

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

# A  Proofs for Main Results

**Proposition 1.** Denote $a^* := \arg\max_{a \in [K]} r(a)$. With constant $\eta > 0$ and any initialization $\theta_1 \in \mathbb{R}^d$, both Algorithms 1 and 2 guarantee $\pi_{\theta_t}^\top r \to r(a^*)$ as $t \to \infty$ on Example 1.

*Proof.* **First part.** Algorithm 1 guarantees $\pi_{\theta_t}^\top r \to r(a^*)$ as $t \to \infty$ on Example 1.

Let $w = (-1, -1)^\top \in \mathbb{R}^d$. We have
$$r' := Xw = (2, 1, -1, -2)^\top, \tag{20}$$
which preserves the ordering of $r \in \mathbb{R}^K$, such that for all $i, j \in [K]$, $r(i) > r(j)$ if and only if $r'(i) > r'(j)$, which means Example 1 satisfies the conditions in Theorem 1. The results then follow by using Theorem 1.

**Second part.** Algorithm 2 guarantees $\pi_{\theta_t}^\top r \to r(a^*)$ as $t \to \infty$ on Example 1.

First, note that $r = (9, 8, 7, 6)^\top$, and $a^* = \arg\max_{a \in [K]} r(a) = 1$. Next, by calculation, we have,
$$\hat{r} := X(X^\top X)^{-1} X^\top r = \frac{1}{5} \cdot (22, -4, -11, 8)^\top. \tag{21}$$
Therefore, we have, $\hat{r}(a^*) = \hat{r}(1) > \hat{r}(a)$ for all $a \neq a^*$, which means Example 1 satisfies the conditions in Theorem 2. The results then follow by using Theorem 2. $\qquad\square$

**Lemma 1** (No stationary points in finite region). Under the same conditions as Theorem 1, and $r(i) \neq r(j)$ for all $i \neq j$ (unique action reward), Algorithm 1 assures $\|\theta_t\|_2 \to \infty$ and $\pi_{\theta_t}(i) \to 1$ for an action $i \in [K]$ as $t \to \infty$.

*Proof.* According to Lemma 3, we have, for all $t \geq 1$,
$$\left| (\pi_{\theta_{t+1}} - \pi_{\theta_t})^\top r - \left\langle \frac{d\,\pi_{\theta_t}^\top r}{d\theta_t}, \theta_{t+1} - \theta_t \right\rangle \right| \leq \frac{9}{4} \cdot \|r\|_\infty \cdot \lambda_{\max}(X^\top X) \cdot \|\theta_{t+1} - \theta_t\|_2^2, \tag{22}$$
which implies that,
$$\pi_{\theta_{t+1}}^\top r - \pi_{\theta_t}^\top r \geq \left\langle \frac{d\,\pi_{\theta_t}^\top r}{d\theta_t}, \theta_{t+1} - \theta_t \right\rangle - \frac{9}{4} \cdot \|r\|_\infty \cdot \lambda_{\max}(X^\top X) \cdot \|\theta_{t+1} - \theta_t\|_2^2 \tag{23}$$
$$= \left( \eta - \eta^2 \cdot \frac{9}{4} \cdot \|r\|_\infty \cdot \lambda_{\max}(X^\top X) \right) \cdot \left\| \frac{d\,\pi_{\theta_t}^\top r}{d\theta_t} \right\|_2^2. \tag{24}$$
Using a constant learning rate,
$$0 < \eta < \frac{4}{9 \cdot \|r\|_\infty \cdot \lambda_{\max}(X^\top X)}, \tag{25}$$
we have,
$$\pi_{\theta_{t+1}}^\top r - \pi_{\theta_t}^\top r \geq \eta \cdot \left( 1 - \eta \cdot \frac{9 \cdot \|r\|_\infty \cdot \lambda_{\max}(X^\top X)}{4} \right) \cdot \left\| \frac{d\,\pi_{\theta_t}^\top r}{d\theta_t} \right\|_2^2 \geq 0. \tag{26}$$
Note that $\pi_{\theta_t}^\top r \leq r(a^*) < \infty$. According to the monotone convergence, $\pi_{\theta_t}^\top r \to c \leq r(a^*)$ as $t \to \infty$. According to Eq. (26), we have,
$$\lim_{t \to \infty} \left\| \frac{d\,\pi_{\theta_t}^\top r}{d\theta_t} \right\|_2^2 = 0. \tag{27}$$
Next, we prove that there is no stationary points in finite region by contradiction. Suppose there exists $\theta' \in \mathbb{R}^d$ ($\|\theta'\|_2 < \infty$), such that,
$$\frac{d\,\pi_{\theta'}^\top r}{d\theta'} = X^\top \left( \mathrm{diag}(\pi_{\theta'}) - \pi_{\theta'} \pi_{\theta'}^\top \right) r = \mathbf{0}. \tag{28}$$
Taking inner product with $w \in \mathbb{R}^K$ on both sides of Eq. (28), we have,
$$w^\top X^\top \left( \mathrm{diag}(\pi_{\theta'}) - \pi_{\theta'} \pi_{\theta'}^\top \right) r = {r'}^\top \left( \mathrm{diag}(\pi_{\theta'}) - \pi_{\theta'} \pi_{\theta'}^\top \right) r \qquad (r' := Xw) \tag{29}$$
$$= w^\top \mathbf{0} = 0. \tag{30}$$

Since $\|\theta'\|_2 < \infty$ and $X$ is bounded ($\max_{i\in[K],\, j\in[d]} |X_{i,j}| \leq C$ for some $C < \infty$), we have, for all $i \in [K]$,

$$\pi_{\theta'}(i) = \frac{\exp\{[X\theta'](i)\}}{\sum_{j\in[K]} \exp\{[X\theta'](j)\}} > 0. \tag{31}$$

Next, according to Lemma 4, we have,

$${r'}^\top \left(\mathrm{diag}(\pi_{\theta'}) - \pi_{\theta'}\pi_{\theta'}^\top\right) r = \sum_{i=1}^{K-1} \pi_{\theta'}(i) \cdot \sum_{j=i+1}^{K} \pi_{\theta'}(j) \cdot (r'(i) - r'(j)) \cdot (r(i) - r(j)). \tag{32}$$

Given any non-trivial reward vector, i.e., $r \neq c \cdot \mathbf{1}$ for any $c \in \mathbb{R}$, since $r' \in \mathbb{R}^K$ preserves the order of $r \in \mathbb{R}^K$, i.e., for all $i, j \in [K]$, $r(i) > r(j)$ iff $r'(i) > r'(j)$, we have, for all $i, j \in [K]$,

$$(r'(i) - r'(j)) \cdot (r(i) - r(j)) \geq 0. \tag{33}$$

On the other hand, since $r \neq c \cdot \mathbf{1}$, there exists at least one pair of $i \neq j$, such that,

$$(r'(i) - r'(j)) \cdot (r(i) - r(j)) > 0. \tag{34}$$

Combining Eqs. (28), (29) and (31) to (34), we have,

$$0 = w^\top \mathbf{0} = w^\top \left(\frac{d\,\pi_{\theta'}^\top r}{d\theta'}\right) \tag{35}$$

$$= w^\top X^\top \left(\mathrm{diag}(\pi_{\theta'}) - \pi_{\theta'}\pi_{\theta'}^\top\right) r \tag{36}$$

$$= {r'}^\top \left(\mathrm{diag}(\pi_{\theta'}) - \pi_{\theta'}\pi_{\theta'}^\top\right) r \tag{37}$$

$$> 0, \tag{38}$$

which is a contradiction. Thus we have, for any $\theta' \in \mathbb{R}^d$ ($\|\theta'\|_2 < \infty$), $\theta'$ is not a stationary point.

Next, we show that $\|\theta_t\|_2 \to \infty$ as $t \to \infty$ also by contradiction. Suppose there exists $C < 0$, such that for all $t \geq 1$,

$$\theta_t \in S_C := \{\theta \in \mathbb{R}^d : \|\theta\|_2 \leq C\}. \tag{39}$$

From the above arguments, we have, for all $\theta \in S_C$, $\left\|\frac{d\,\pi_\theta^\top r}{d\theta}\right\|_2 > 0$. Since $S_C$ is compact, we have,

$$\inf_{\theta \in S_C} \left\|\frac{d\,\pi_\theta^\top r}{d\theta}\right\|_2 \geq \varepsilon > 0, \tag{40}$$

for some $\varepsilon > 0$, which implies that, for all $t \geq 1$,

$$\left\|\frac{d\,\pi_{\theta_t}^\top r}{d\theta_t}\right\|_2 \geq \varepsilon > 0, \tag{41}$$

contradicting Eq. (27). Therefore, we have, $\|\theta_t\|_2 \to \infty$ as $t \to \infty$.

Next, we show that $\pi_{\theta_t}(i) \to 1$ for an action $i \in [K]$ as $t \to \infty$. Suppose $\pi_{\theta_t}(i) \not\to 1$ for any action $i \in [K]$, then there exists at least two different actions $j \neq k$ such that $\pi_{\theta_t}(j) \not\to 0$ and $\pi_{\theta_t}(k) \not\to 0$. Using similar calculations in Eq. (32), we have, $\left\|\frac{d\,\pi_{\theta_t}^\top r}{d\theta_t}\right\|_2 \not\to 0$ as $t \to \infty$, contradicting Eq. (27). Therefore, $\pi_{\theta_t}(i) \to 1$ for an action $i \in [K]$ as $t \to \infty$, i.e. $\pi_{\theta_t}$ approaches a one-hot policy. $\qquad\square$

**Lemma 2** (Non-negative co-variance of order preservation). If $r' \in \mathbb{R}^K$ preserves the order of $r \in \mathbb{R}^K$, i.e., for all $i, j \in [K]$, $r(i) > r(j)$ if and only if $r'(i) > r'(j)$, then for any policy $\pi \in \Delta(K)$,

$${r'}^\top \left(\mathrm{diag}(\pi) - \pi\pi^\top\right) r = \mathrm{Cov}_\pi\left(r', r\right) \geq 0. \tag{42}$$

*Proof.* According to Lemma 4, we have, for all policy $\pi \in \Delta(K)$,

$${r'}^\top \left(\mathrm{diag}(\pi) - \pi\pi^\top\right) r = \sum_{i=1}^{K-1} \pi(i) \cdot \sum_{j=i+1}^{K} \pi(j) \cdot (r'(i) - r'(j)) \cdot (r(i) - r(j)). \tag{43}$$

Since $r' \in \mathbb{R}^K$ preserves the order of $r \in \mathbb{R}^K$, i.e., for all $i, j \in [K]$, $r(i) > r(j)$ if and only if $r'(i) > r'(j)$, we have, for all $i \neq j$,

$$(r'(i) - r'(j)) \cdot (r(i) - r(j)) \geq 0. \tag{44}$$

Combining Eqs. (43) and (44), we have Eq. (42). $\qquad\square$

**Theorem 1** (Reward order preservation, non-domination features). Given any reward $r \in \mathbb{R}^K$ and feature matrix $X \in \mathbb{R}^{K \times d}$. Denote $x_i \in \mathbb{R}^d$ as the $i$-th row vector of $X$. If **(i)** $x_i^\top x_i > x_i^\top x_j$ for all $j \neq i$, and **(ii)** there exists at least one $w \in \mathbb{R}^d$, s.t., $r' := Xw$ preserves the order of $r$, i.e., for all $i, j \in [K]$, $r(i) > r(j)$ if and only if $r'(i) > r'(j)$, then for any initialization $\theta_1 \in \mathbb{R}^d$, Algorithm 1 with a constant learning rate $\eta > 0$ achieves global convergence of $\pi_{\theta_t}^\top r \to r(a^*)$ as $t \to \infty$.

*Proof.* **First part.** According to Lemma 1, using any constant learning rate,

$$0 < \eta < \frac{4}{9 \cdot \|r\|_\infty \cdot \lambda_{\max}(X^\top X)}, \tag{45}$$

Algorithm 1 guarantees that $\|\theta_t\|_2 \to \infty$ as $t \to \infty$, and $\pi_{\theta_t}(i) \to 1$ for an action $i \in [K]$ as $t \to \infty$.

**Second part.** For the direction $w \in \mathbb{R}^d$ such that $r' := Xw$ preserves the order of $r$. We have,

$$w^\top \theta_{t+1} = w^\top \theta_t + \eta \cdot w^\top X^\top \left( \mathrm{diag}(\pi_{\theta_t}) - \pi_{\theta_t} \pi_{\theta_t}^\top \right) r \qquad \text{(by Algorithm 1)} \tag{46}$$

$$= w^\top \theta_t + \eta \cdot {r'}^\top \left( \mathrm{diag}(\pi_{\theta_t}) - \pi_{\theta_t} \pi_{\theta_t}^\top \right) r \qquad (r' := Xw) \tag{47}$$

$$\geq w^\top \theta_t. \qquad \text{(by Lemma 2)} \tag{48}$$

**Third part.** Suppose there exists a sub-optimal action $i \in [K]$ with $r(i) < r(a^*)$, and $\pi_{\theta_t}(i) \to 1$ as $t \to \infty$. Then we have, for all large enough $t \geq 1$,

$$\pi_{\theta_t}(i) > \pi_{\theta_t}(j), \tag{49}$$

for all $j \neq i$, which implies that,

$$\left[ \frac{X\theta_t}{\|\theta_t\|_2} \right](i) = \max_{a \in [K]} \left[ \frac{X\theta_t}{\|\theta_t\|_2} \right](a). \tag{50}$$

Now we prove by contradiction that the assumption of $\pi_{\theta_t}(i) \to 1$ as $t \to \infty$ cannot be true for any sub-optimal action $i \in [K]$ with $r(i) < r(a^*)$.

Using the sub-optimal action's reward $r(i)$, the action set $[K]$ can be partitioned as follows,

$$\mathcal{A}(i) := \{ j \in [K] : r(j) = r(i) \}, \tag{51}$$

$$\mathcal{A}^+(i) := \{ a^+ \in [K] : r(a^+) > r(i) \}, \tag{52}$$

$$\mathcal{A}^-(i) := \{ a^- \in [K] : r(a^-) < r(i) \}. \tag{53}$$

According to Eq. (50), for all large enough $t \geq 1$, $i = \arg\max_{a \in [K]} [X\theta_t](a)$. Take the second largest component of $X\theta_t$, and denote the corresponding action index as $j$.

**Case 1.** $j \in \mathcal{A}^+(i)$. This means $j = a^+$ for a "good" action with $r(a^+) > r(i)$.

We have, for all large enough $t \geq 1$, for all "bad" action $a^- \in \mathcal{A}^-(i)$,

$$[X\theta_t](a^+) - [X\theta_t](a^-) = \|\theta_t\|_2 \cdot \left( \left[ \frac{X\theta_t}{\|\theta_t\|_2} \right](a^+) - \left[ \frac{X\theta_t}{\|\theta_t\|_2} \right](a^-) \right) \tag{54}$$

$$\geq c \cdot \|\theta_t\|_2, \tag{55}$$

for some $c > 0$ according to $j = a^+$. Next, we have,

$$\frac{\pi_{\theta_t}(a^+)}{\pi_{\theta_t}(a^-)} = \exp\left\{ [X\theta_t](a^+) - [X\theta_t](a^-) \right\} \geq \exp\left\{ c \cdot \|\theta_t\|_2 \right\}, \tag{56}$$

which implies that,

$$r(i) - \pi_{\theta_t}^\top r = \sum_{k \neq i} \pi_{\theta_t}(k) \cdot (r(i) - r(k)) \tag{57}$$

$$= - \sum_{\tilde{a}^+ \in \mathcal{A}^+(i)} \pi_{\theta_t}(\tilde{a}^+) \cdot \left( r(\tilde{a}^+) - r(i) \right) + \sum_{a^- \in \mathcal{A}^-(i)} \pi_{\theta_t}(a^-) \cdot \left( r(i) - r(a^-) \right) \tag{58}$$

$$\leq -\pi_{\theta_t}(a^+) \cdot \left( r(a^+) - r(i) \right) + \sum_{a^- \in \mathcal{A}^-(i)} \pi_{\theta_t}(a^-) \cdot \left( r(i) - r(a^-) \right) \tag{59}$$

$$= -\pi_{\theta_t}(a^+) \cdot \left[ \underbrace{r(a^+) - r(i)}_{>0} - \sum_{a^- \in \mathcal{A}^-(i)} \underbrace{\frac{\pi_{\theta_t}(a^-)}{\pi_{\theta_t}(a^+)}}_{\to 0} \cdot \left( r(i) - r(a^-) \right) \right] \tag{60}$$

$$< 0, \tag{61}$$

where $r(a^+) - r(i) > 0$ is from Eq. (52), $\frac{\pi_{\theta_t}(a^-)}{\pi_{\theta_t}(a^+)} \to 0$ as $t \to \infty$ is by Eq. (56) and Lemma 1. Eq. (57) means that $\pi_{\theta_t}^\top r > r(i)$ happens at a finite time $t < \infty$. According to Eq. (26), we have, for all large enough $t \geq 1$, $\pi_{\theta_t}^\top r > r(i)$, which is a contradiction with $\pi_{\theta_t}^\top r \to r(i)$ implied by the assumption of $\pi_{\theta_t}(i) \to 1$ as $t \to \infty$.

**Case 2.** $j \in \mathcal{A}^-(i)$. This means $j = a^-$ for a "bad" action $r(a^-) < r(i)$.

Using similar arguments around Eq. (56), we have, for all $a \in [K]$ such that $a \neq i$ and $a \neq a^-$,

$$\frac{\pi_{\theta_t}(a^-)}{\pi_{\theta_t}(a)} = \exp\left\{[X\theta_t](a^-) - [X\theta_t](a)\right\} \geq \exp\left\{c \cdot \|\theta_t\|_2\right\}, \tag{62}$$

for some $c > 0$. Consider a direction $u \in \mathbb{R}^d$, $\|u\|_2 = 1$, such that,

$$[Xu](a^-) = \max_{a \in [K]}[Xu](a). \tag{63}$$

According to Algorithm 1, we have,

$$u^\top \theta_{t+1} = u^\top \theta_t + \eta \cdot u^\top X^\top \left(\mathrm{diag}(\pi_{\theta_t}) - \pi_{\theta_t}\pi_{\theta_t}^\top\right) r \tag{64}$$

$$= u^\top \theta_t + \eta \cdot u^\top X^\top \left(\mathrm{diag}(\pi_{\theta_t}) - \pi_{\theta_t}\pi_{\theta_t}^\top\right)(r - r(i) \cdot \mathbf{1}), \tag{65}$$

where the last equation is because of,

$$\left(\mathrm{diag}(\pi_{\theta_t}) - \pi_{\theta_t}\pi_{\theta_t}^\top\right) \mathbf{1} = \pi_{\theta_t} - \pi_{\theta_t} \cdot (\pi_{\theta_t}^\top \mathbf{1}) = \pi_{\theta_t} - \pi_{\theta_t} = \mathbf{0}. \tag{66}$$

Denote $y := Xu$. We have,

$$\left(\mathrm{diag}(\pi_{\theta_t}) - \pi_{\theta_t}\pi_{\theta_t}^\top\right) Xu = \begin{bmatrix} \pi_{\theta_t}(1) \cdot \left(y(1) - \pi_{\theta_t}^\top y\right) \\ \pi_{\theta_t}(2) \cdot \left(y(2) - \pi_{\theta_t}^\top y\right) \\ \vdots \\ \pi_{\theta_t}(K) \cdot \left(y(K) - \pi_{\theta_t}^\top y\right) \end{bmatrix} \in \mathbb{R}^K. \tag{67}$$

Therefore, from Eqs. (64) and (67), we have,

$$u^\top X^\top \left(\mathrm{diag}(\pi_{\theta_t}) - \pi_{\theta_t}\pi_{\theta_t}^\top\right) r = \sum_{a \neq i} \pi_{\theta_t}(a) \cdot \left(y(a) - \pi_{\theta_t}^\top y\right) \cdot (r(a) - r(i)) \tag{68}$$

$$= \sum_{a \neq i,\, a \neq a^-} \pi_{\theta_t}(a) \cdot \left(y(a) - \pi_{\theta_t}^\top y\right) \cdot (r(a) - r(i)) + \pi_{\theta_t}(a^-) \cdot \left(y(a^-) - \pi_{\theta_t}^\top y\right) \cdot \left(r(a^-) - r(i)\right) \tag{69}$$

$$= -\pi_{\theta_t}(a^-) \cdot \left[\underbrace{\left(y(a^-) - \pi_{\theta_t}^\top y\right)}_{>0} \cdot \underbrace{\left(r(i) - r(a^-)\right)}_{>0} - \sum_{\substack{a \neq i, \\ a \neq a^-}} \underbrace{\frac{\pi_{\theta_t}(a)}{\pi_{\theta_t}(a^-)}}_{\to 0} \cdot \underbrace{\left(y(a) - \pi_{\theta_t}^\top y\right)}_{\text{bounded}} \cdot (r(a) - r(i))\right] \tag{70}$$

$$< 0, \tag{71}$$

where $y(a^-) - \pi_{\theta_t}^\top y > 0$ is by Eq. (63), $r(i) - r(a^-) > 0$ is from Eq. (53), $\frac{\pi_{\theta_t}(a)}{\pi_{\theta_t}(a^-)} \to 0$ as $t \to \infty$ is according to Eq. (62) and Lemma 1, and $y(a) - \pi_{\theta_t}^\top y$ is bounded is because of $X$ and $u$ are bounded $(\max_{i \in [K],\, j \in [d]} |X_{i,j}| \leq C$, and $\max_{j \in [d]} |u(j)| \leq C$ for some $C < \infty)$.

Combining Eqs. (64) and (68), we have, for all large enough $t \geq 1$,

$$u^\top \theta_{t+1} = u^\top \theta_t + \eta \cdot u^\top X^\top \left(\mathrm{diag}(\pi_{\theta_t}) - \pi_{\theta_t}\pi_{\theta_t}^\top\right) r \tag{72}$$

$$< u^\top \theta_t, \tag{73}$$

which implies that $u^\top \theta_t \to -\infty$ as $t \to \infty$ according to Lemma 1. On the other hand, according to Eq. (46), we have, for all large enough $t \geq 1$,

$$w^\top \theta_{t+1} > w^\top \theta_t, \tag{74}$$

which implies that $w^\top \theta_t \to \infty$ as $t \to \infty$ according to Lemma 1. According to the non-domination feature condition, i.e., $x_i^\top x_i > x_i^\top x_j$ for all $i \neq j$, we have, for any "bad" action $a^- \in \mathcal{A}^-(i)$,

$$[Xx_{a^-}](a^-) = \max_{a \in [K]}[Xx_{a^-}](a), \tag{75}$$

which implies that,
$$[X\theta_{t+1}](a^-) = x_{a^-}^\top \theta_{t+1} < x_{a^-}^\top \theta_t = [X\theta_t](a^-), \tag{76}$$
by taking $u = x_{a^-}$ in Eq. (72). This means the score of a "bad" action $a^- \in \mathcal{A}^-(i)$ is monotonically decreasing, and it approaches $-\infty$ due to Lemma 1. On the other hand, take $w = x_{a^*}$ in Eq. (74) (or take $u = -x_{a^*}$ in Eq. (72)), we have,
$$[X\theta_{t+1}](a^*) = x_{a^*}^\top \theta_{t+1} > x_{a^*}^\top \theta_t = [X\theta_t](a^*), \tag{77}$$
which means the optimal action's score is monotonically increasing, it approaches $\infty$ due to Lemma 1. Therefore, we have, for all large enough $t \geq 1$,
$$[X\theta_t](a^*) > [X\theta_t](a^-), \tag{78}$$
for any "bad" action $a^- \in \mathcal{A}^-(i)$, contradicting the assumption of $j = a^- \in \mathcal{A}^-(i)$. $\qquad\square$

**Proposition 2.** Let $K = 3$, $d = 2$, $X^\top = \begin{bmatrix} 0 & -10 & 0 \\ -2 & 4 & 1 \end{bmatrix} \in \mathbb{R}^{d \times K}$, and $r = Xw = (4, 2, -2)^\top$, where $w = (-1, -2)^\top \in \mathbb{R}^d$. With initialization $\theta_1 = (-\ln 2, \ln 2)^\top$, Algorithm 1 does not achieve global convergence, i.e., $\pi_{\theta_t}(1) \not\to 1$ as $t \to \infty$.

*Proof.* Define the following region,
$$\mathcal{R} := \left\{ \theta \in \mathbb{R}^d : \frac{\pi_\theta(1)}{\pi_\theta(3)} < \frac{1}{2}, \text{ and } \frac{\pi_\theta(2)}{\pi_\theta(3)} > 3 \right\}. \tag{79}$$
We show that, **(i)** $\theta_1 = (-\ln 2, \ln 2)^\top \in \mathcal{R}$, and **(ii)** if $\theta_t \in \mathcal{R}$, then $\theta_{t+1} \in \mathcal{R}$, and,
$$\frac{\pi_{\theta_{t+1}}(1)}{\pi_{\theta_{t+1}}(3)} < \frac{\pi_{\theta_t}(1)}{\pi_{\theta_t}(3)}, \tag{80}$$
which means that, for all $t \geq 1$, we have, $\frac{\pi_{\theta_t}(1)}{\pi_{\theta_t}(3)} < 1/2$, implying that $\pi_{\theta_t}(1) \not\to 1$ as $t \to \infty$.

**First part. (i)** $\theta_1 = (-\ln 2, \ln 2)^\top \in \mathcal{R}$.

For the initialization $\theta_1 = (-\ln 2, \ln 2)^\top$, we have,
$$\exp\{X\theta_1\} = (2^{-2}, 2^{14}, 2)^\top. \tag{81}$$
By calculation, we have,
$$\frac{\pi_{\theta_1}(1)}{\pi_{\theta_1}(3)} = \frac{\exp\{[X\theta_1](1)\}}{\exp\{[X\theta_1](3)\}} = \frac{1}{8} < \frac{1}{4} \cdot \frac{r(2) - r(3)}{r(1) - r(2)} = \frac{1}{2}, \text{ and} \tag{82}$$
$$\frac{\pi_{\theta_1}(2)}{\pi_{\theta_1}(3)} = \frac{\exp\{[X\theta_1](2)\}}{\exp\{[X\theta_1](3)\}} = 2^{13} > \frac{r(1) - r(3)}{r(1) - r(2)} = 3, \tag{83}$$
which verifies that $\theta_1 = (-\ln 2, \ln 2)^\top \in \mathcal{R}$.

**Second part. (ii)** If $\theta_t \in \mathcal{R}$, then $\theta_{t+1} \in \mathcal{R}$, and $\frac{\pi_{\theta_{t+1}}(1)}{\pi_{\theta_{t+1}}(3)} < \frac{\pi_{\theta_t}(1)}{\pi_{\theta_t}(3)}$.

Suppose $\theta_t \in \mathcal{R}$, we have,
$$\frac{\pi_{\theta_t}(1)}{\pi_{\theta_t}(3)} < \frac{1}{2}, \text{ and } \frac{\pi_{\theta_t}(2)}{\pi_{\theta_t}(3)} > 3. \tag{84}$$
Next, we have,
$$r(2) - \pi_{\theta_t}^\top r = \pi_{\theta_t}(1) \cdot (r(2) - r(1)) + \pi_{\theta_t}(3) \cdot (r(2) - r(3)) \tag{85}$$
$$= -2 \cdot \pi_{\theta_t}(1) + 4 \cdot \pi_{\theta_t}(3) \tag{86}$$
$$= 2 \cdot \pi_{\theta_t}(3) \cdot \left( -\frac{\pi_{\theta_t}(1)}{\pi_{\theta_t}(3)} + 2 \right) > 0, \tag{87}$$

and

$$\frac{r(1) - \pi_{\theta_t}^\top r}{\pi_{\theta_t}^\top r - r(3)} = \frac{\pi_{\theta_t}(2) \cdot (r(1) - r(2)) + \pi_{\theta_t}(3) \cdot (r(1) - r(3))}{\pi_{\theta_t}(1) \cdot (r(1) - r(3)) + \pi_{\theta_t}(2) \cdot (r(2) - r(3))} \tag{88}$$

$$= \frac{2 \cdot \pi_{\theta_t}(2) + 6 \cdot \pi_{\theta_t}(3)}{6 \cdot \pi_{\theta_t}(1) + 4 \cdot \pi_{\theta_t}(2)} \tag{89}$$

$$< \frac{2 \cdot \pi_{\theta_t}(2) + 6 \cdot \pi_{\theta_t}(3)}{4 \cdot \pi_{\theta_t}(2)} \tag{90}$$

$$= \frac{1}{2} + \frac{3}{2} \cdot \frac{\pi_{\theta_t}(3)}{\pi_{\theta_t}(2)} < 1. \tag{91}$$

According to Algorithm 1, we have,

$$\theta_{t+1} - \theta_t = \eta \cdot X^\top \left( \text{diag}(\pi_{\theta_t}) - \pi_{\theta_t} \pi_{\theta_t}^\top \right) r \tag{92}$$

$$= \eta \cdot \begin{bmatrix} 0 & -10 & 0 \\ -2 & 4 & 1 \end{bmatrix} \begin{bmatrix} \pi_{\theta_t}(1) \cdot \left( r(1) - \pi_{\theta_t}^\top r \right) \\ \pi_{\theta_t}(2) \cdot \left( r(2) - \pi_{\theta_t}^\top r \right) \\ \pi_{\theta_t}(3) \cdot \left( r(3) - \pi_{\theta_t}^\top r \right) \end{bmatrix} \tag{93}$$

$$= \eta \cdot \begin{bmatrix} -10 \cdot \pi_{\theta_t}(2) \cdot \left( r(2) - \pi_{\theta_t}^\top r \right) \\ -2 \cdot \pi_{\theta_t}(1) \cdot \left( r(1) - \pi_{\theta_t}^\top r \right) + 4 \cdot \pi_{\theta_t}(2) \cdot \left( r(2) - \pi_{\theta_t}^\top r \right) + \pi_{\theta_t}(3) \cdot \left( r(3) - \pi_{\theta_t}^\top r \right) \end{bmatrix}. \tag{94}$$

Next, we have,

$$- 2 \cdot \pi_{\theta_t}(1) \cdot \left( r(1) - \pi_{\theta_t}^\top r \right) + 4 \cdot \pi_{\theta_t}(2) \cdot \left( r(2) - \pi_{\theta_t}^\top r \right) + \pi_{\theta_t}(3) \cdot \left( r(3) - \pi_{\theta_t}^\top r \right) \tag{95}$$

$$= -6 \cdot \pi_{\theta_t}(1) \cdot \left( r(1) - \pi_{\theta_t}^\top r \right) - 3 \cdot \pi_{\theta_t}(3) \cdot \left( r(3) - \pi_{\theta_t}^\top r \right) \tag{96}$$

$$= 3 \cdot \pi_{\theta_t}(3) \cdot \left( \pi_{\theta_t}^\top r - r(3) \right) \cdot \left[ -2 \cdot \frac{\pi_{\theta_t}(1)}{\pi_{\theta_t}(3)} \cdot \frac{r(1) - \pi_{\theta_t}^\top r}{\pi_{\theta_t}^\top r - r(3)} + 1 \right] \tag{97}$$

$$> 3 \cdot \pi_{\theta_t}(3) \cdot \left( \pi_{\theta_t}^\top r - r(3) \right) \cdot \left( -2 \cdot \frac{1}{2} \cdot 1 + 1 \right) = 0, \tag{98}$$

which implies that,

$$\theta_{t+1}(2) > \theta_t(2). \tag{99}$$

Therefore, we have,

$$\frac{\pi_{\theta_{t+1}}(1)}{\pi_{\theta_{t+1}}(3)} = \frac{\exp\{[X\theta_{t+1}](1)\}}{\exp\{[X\theta_{t+1}](3)\}} = \frac{\exp\{-2 \cdot \theta_{t+1}(2)\}}{\exp\{\theta_{t+1}(2)\}} < \frac{\exp\{-2 \cdot \theta_t(2)\}}{\exp\{\theta_t(2)\}} = \frac{\pi_{\theta_t}(1)}{\pi_{\theta_t}(3)} < \frac{1}{2}. \tag{100}$$

On the other hand, we have,

$$-10 \cdot \pi_{\theta_t}(2) \cdot \left( r(2) - \pi_{\theta_t}^\top r \right) < 0, \tag{101}$$

which implies that,

$$\theta_{t+1}(1) < \theta_t(1). \tag{102}$$

Therefore, we have,

$$\frac{\pi_{\theta_{t+1}}(2)}{\pi_{\theta_{t+1}}(3)} = \frac{\exp\{[X\theta_{t+1}](2)\}}{\exp\{[X\theta_{t+1}](3)\}} = \frac{\exp\{-10 \cdot \theta_{t+1}(1) + 4 \cdot \theta_{t+1}(2)\}}{\exp\{\theta_{t+1}(2)\}} \tag{103}$$

$$> \frac{\exp\{[X\theta_{t+1}](2)\}}{\exp\{[X\theta_{t+1}](3)\}} = \frac{\exp\{-10 \cdot \theta_t(1) + 4 \cdot \theta_t(2)\}}{\exp\{\theta_t(2)\}} \tag{104}$$

$$= \frac{\pi_{\theta_t}(2)}{\pi_{\theta_t}(3)} > 3, \tag{105}$$

which proves that $\theta_{t+1} \in \mathcal{R}$ and $\frac{\pi_{\theta_{t+1}}(1)}{\pi_{\theta_{t+1}}(3)} < \frac{\pi_{\theta_t}(1)}{\pi_{\theta_t}(3)}$. $\qquad\square$

**Theorem 2** (Optimal action preservation condition). For a constant learning rate $\eta > 0$, a necessary and sufficient condition for Algorithm 2 to achieve global convergence $\pi_{\theta_t}^\top r \to r(a^*)$ as $t \to \infty$ from any initialization $\theta_1 \in \mathbb{R}^d$ is that $\hat{r}(a^*) > \hat{r}(a)$ for all $a \neq a^*$, such that $a^* := \arg\max_{a \in [K]} r(a)$, and $\hat{r} := X \left( X^\top X \right)^{-1} X^\top r$ is the least squares projection of $r$ onto the column space of $X$. If the condition is satisfied, then the rate of convergence is $(\pi^* - \pi_{\theta_t})^\top r \in O(e^{-c \cdot t})$ for some $c > 0$.

*Proof.* **First part.** Sufficiency. Suppose that $\hat{r}(a^*) > \hat{r}(a)$ for all $a \neq a^*$. Denote

$$\hat{\Delta} := \hat{r}(a^*) - \max_{a \neq a^*} \hat{r}(a). \tag{106}$$

According to Algorithm 2, we have, for all $t \geq 1$,

$$X\theta_{t+1} = X\theta_t + \eta \cdot X \left( X^\top X \right)^{-1} X^\top r = X\theta_t + \eta \cdot \hat{r}. \tag{107}$$

Next, we have, for all $a \neq a^*$,

$$\frac{\pi_{\theta_{t+1}}(a^*)}{\pi_{\theta_{t+1}}(a)} = \exp \left\{ [X\theta_{t+1}](a^*) - [X\theta_{t+1}](a) \right\} \qquad \text{(by Eq. (3))} \tag{108}$$

$$= \exp \left\{ [X\theta_t](a^*) - [X\theta_t](a) + \eta \cdot (\hat{r}(a^*) - \hat{r}(a)) \right\} \qquad \text{(by Eq. (107))} \tag{109}$$

$$= \exp \left\{ [X\theta_1](a^*) - [X\theta_1](a) + \eta \cdot (\hat{r}(a^*) - \hat{r}(a)) \cdot t \right\} \tag{110}$$

$$\geq \exp \left\{ [X\theta_1](a^*) - [X\theta_1](a) + \eta \cdot \hat{\Delta} \cdot t \right\} \qquad \text{(by Eq. (106))} \tag{111}$$

$$= \frac{\pi_{\theta_1}(a^*)}{\pi_{\theta_1}(a)} \cdot e^{\eta \cdot \hat{\Delta} \cdot t}, \tag{112}$$

which implies that,

$$\frac{1}{\pi_{\theta_t}(a^*)} - 1 = \sum_{a \neq a^*} \frac{\pi_{\theta_t}(a)}{\pi_{\theta_t}(a^*)} \tag{113}$$

$$\leq \sum_{a \neq a^*} \frac{\pi_{\theta_1}(a)}{\pi_{\theta_1}(a^*)} \cdot e^{-\eta \cdot \hat{\Delta} \cdot (t-1)} \qquad \text{(by Eq. (108))} \tag{114}$$

$$\leq c(X, \theta_1) \cdot K \cdot e^{-\eta \cdot \hat{\Delta} \cdot (t-1)}, \tag{115}$$

where

$$c(X, \theta_1) := \max_{a \neq a^*} \frac{\pi_{\theta_1}(a)}{\pi_{\theta_1}(a^*)}. \tag{116}$$

Therefore, we have,

$$\left( \pi^* - \pi_{\theta_t} \right)^\top r = \sum_{a \neq a^*} \pi_{\theta_t}(a) \cdot (r(a^*) - r(a)) \tag{117}$$

$$\leq 2 \cdot \|r\|_\infty \cdot (1 - \pi_{\theta_t}(a^*)) \qquad \left( \text{using } r \in \left[ -\|r\|_\infty, \|r\|_\infty \right]^K \right) \tag{118}$$

$$= 2 \cdot \|r\|_\infty \cdot \left( 1 - \frac{1}{\frac{1}{\pi_{\theta_t}(a^*)} - 1 + 1} \right) \tag{119}$$

$$\leq 2 \cdot \|r\|_\infty \cdot \left( 1 - \frac{1}{c(X, \theta_1) \cdot K \cdot e^{-\eta \cdot \hat{\Delta} \cdot (t-1)} + 1} \right) \qquad \text{(by Eq. (113))} \tag{120}$$

$$= \frac{2 \cdot \|r\|_\infty \cdot c(X, \theta_1) \cdot K}{c(X, \theta_1) \cdot K + \exp \left\{ \eta \cdot \hat{\Delta} \cdot (t-1) \right\}}, \tag{121}$$

which proves the sufficiency and the convergence rate $\left( \pi^* - \pi_{\theta_t} \right)^\top r \in O(e^{-c \cdot t})$ for some $c > 0$.

**Second part.** Necessity. Suppose the condition is not satisfied, i.e., there exists one sub-optimal action $a \neq a^*$, such that $\hat{r}(a^*) \leq \hat{r}(a)$. We have, for all $t \geq 1$,

$$\frac{\pi_{\theta_{t+1}}(a^*)}{\pi_{\theta_{t+1}}(a)} = \exp \left\{ [X\theta_{t+1}](a^*) - [X\theta_{t+1}](a) \right\} \qquad \text{(by Eq. (3))} \tag{122}$$

$$= \exp \left\{ [X\theta_t](a^*) - [X\theta_t](a) + \eta \cdot (\hat{r}(a^*) - \hat{r}(a)) \right\} \qquad \text{(by Eq. (107))} \tag{123}$$

$$\leq \exp \left\{ [X\theta_1](a^*) - [X\theta_1](a) + \eta \cdot (\hat{r}(a^*) - \hat{r}(a)) \cdot t \right\} \tag{124}$$

$$\leq \frac{\pi_{\theta_1}(a^*)}{\pi_{\theta_1}(a)}, \qquad \text{(using } \hat{r}(a^*) \leq \hat{r}(a)) \tag{125}$$

which implies that,

$$\frac{1}{\pi_{\theta_t}(a^*)} - 1 = \sum_{a' \neq a^*} \frac{\pi_{\theta_t}(a')}{\pi_{\theta_t}(a^*)} \tag{126}$$

$$\geq \frac{\pi_{\theta_t}(a)}{\pi_{\theta_t}(a^*)} \qquad (\pi_{\theta_t}(a') > 0 \text{ for all } a' \in [K]) \tag{127}$$

$$\geq \frac{\pi_{\theta_1}(a)}{\pi_{\theta_1}(a^*)}. \qquad \text{(by Eq. (122))} \tag{128}$$

Therefore, we have,

$$(\pi^* - \pi_{\theta_t})^\top r = \sum_{a \neq a^*} \pi_{\theta_t}(a) \cdot (r(a^*) - r(a)) \tag{129}$$

$$\geq \Delta \cdot (1 - \pi_{\theta_t}(a^*)) \qquad \left(\Delta := r(a^*) - \max_{a \neq a^*} r(a)\right) \tag{130}$$

$$= \Delta \cdot \left(1 - \frac{1}{\frac{1}{\pi_{\theta_t}(a^*)} - 1 + 1}\right) \tag{131}$$

$$\geq \Delta \cdot \left(1 - \frac{1}{\frac{\pi_{\theta_1}(a)}{\pi_{\theta_1}(a^*)} + 1}\right) \qquad \text{(by Eq. (126))} \tag{132}$$

$$= \frac{\Delta \cdot \pi_{\theta_1}(a)}{\pi_{\theta_1}(a) + \pi_{\theta_1}(a^*)} > 0, \tag{133}$$

i.e., $\pi_{\theta_t}^\top r \not\to r(a^*)$ as $t \to \infty$, which proves the necessity of the condition. $\qquad \square$

## B  Miscellaneous Extra Supporting Results

**Lemma 3** (Smoothness). *Given any reward vector $r \in \mathbb{R}^K$ and feature matrix $X \in \mathbb{R}^{K \times d}$. The expected reward function $\theta \mapsto \pi_\theta^\top r$ with $\pi_\theta = \mathrm{softmax}(X\theta)$ is $\beta$-smooth with*

$$\beta = \frac{9}{2} \cdot \|r\|_\infty \cdot \lambda_{\max}(X^\top X), \tag{134}$$

*i.e., for all $\theta$, $\theta' \in \mathbb{R}^d$,*

$$\left|(\pi_{\theta'} - \pi_\theta)^\top r - \left\langle \frac{d\pi_\theta^\top r}{d\theta}, \theta' - \theta \right\rangle\right| \leq \frac{9}{4} \cdot \|r\|_\infty \cdot \lambda_{\max}(X^\top X) \cdot \|\theta' - \theta\|_2^2. \tag{135}$$

*Proof.* Let $S := S(X, r, \theta) \in \mathbb{R}^{d \times d}$ be the second-order derivative of the value map $\theta \mapsto \pi_\theta^\top r$. By Taylor's theorem, it suffices to show that the spectral radius of $S$ (regardless of $\theta$) is bounded by $\beta$. Now, by its definition we have

$$S = \frac{d}{d\theta} \left\{ \frac{d\pi_\theta^\top r}{d\theta} \right\} \tag{136}$$

$$= \frac{d}{d\theta} \left\{ X^\top (\mathrm{diag}(\pi_\theta) - \pi_\theta \pi_\theta^\top) r \right\}. \qquad \text{(by Eq. (4))} \tag{137}$$

Continuing with our calculation fix $i, j \in [d]$. Then,

$$S_{i,j} = \frac{d\left\{ \sum_{a=1}^K X_{a,i} \cdot \pi_\theta(a) \cdot (r(a) - \pi_\theta^\top r) \right\}}{d\theta(j)} \tag{138}$$

$$= \sum_{a=1}^K X_{a,i} \cdot \frac{d\pi_\theta(a)}{d\theta(j)} \cdot (r(a) - \pi_\theta^\top r) - \sum_{a=1}^K X_{a,i} \cdot \pi_\theta(a) \cdot \sum_{a'=1}^K \frac{d\pi_\theta(a')}{d\theta(j)} \cdot r(a'). \tag{139}$$

We have, for all $a \in [K]$ and $j \in [d]$,

$$\frac{d\pi_\theta(a)}{d\theta(j)} = \frac{d}{d\theta(j)}\left\{\frac{\exp\{[X\theta](a)\}}{\sum_{a' \in [K]} \exp\{[X\theta](a')\}}\right\} \tag{140}$$

$$= \frac{\frac{d \exp\{[X\theta](a)\}}{d\theta(j)} \cdot \sum_{a' \in [K]} \exp\{[X\theta](a')\} - \exp\{[X\theta](a)\} \cdot \frac{d \sum_{a' \in [K]} \exp\{[X\theta](a')\}}{d\theta(j)}}{\left(\sum_{a' \in [K]} \exp\{[X\theta](a')\}\right)^2} \tag{141}$$

$$= \frac{\exp\{[X\theta](a)\} \cdot X_{a,j} \cdot \sum_{a' \in [K]} \exp\{[X\theta](a')\} - \exp\{[X\theta](a)\} \cdot \sum_{a' \in [K]} \exp\{[X\theta](a')\} \cdot X_{a',j}}{\left(\sum_{a' \in [K]} \exp\{[X\theta](a')\}\right)^2} \tag{142}$$

$$= \frac{\exp\{[X\theta](a)\} \cdot X_{a,j} - \exp\{[X\theta](a)\} \cdot \sum_{a' \in [K]} \pi_\theta(a') \cdot X_{a',j}}{\sum_{a' \in [K]} \exp\{[X\theta](a')\}} \tag{143}$$

$$= \pi_\theta(a) \cdot \left(X_{a,j} - \sum_{a' \in [K]} \pi_\theta(a') \cdot X_{a',j}\right). \tag{144}$$

Combining Eqs. (138) and (140), we have,

$$S_{i,j} = \sum_{a=1}^{K} X_{a,i} \cdot \pi_\theta(a) \cdot (r(a) - \pi_\theta^\top r) \cdot X_{a,j} - \sum_{a=1}^{K} X_{a,i} \cdot \pi_\theta(a) \cdot (r(a) - \pi_\theta^\top r) \cdot \sum_{a'=1}^{K} \pi_\theta(a') \cdot X_{a',j} \tag{145}$$

$$- \sum_{a=1}^{K} X_{a,i} \cdot \pi_\theta(a) \cdot \sum_{a'=1}^{K} \pi_\theta(a') \cdot \left(X_{a',j} - \sum_{a''=1}^{K} \pi_\theta(a'') \cdot X_{a'',j}\right) \cdot r(a'). \tag{146}$$

To show the bound on the spectral radius of $S$, pick $y \in \mathbb{R}^d$. Then,

$$|y^\top S y| = \left|\sum_{i=1}^{d} \sum_{j=1}^{d} S_{i,j} \cdot y(i) \cdot y(j)\right| \tag{147}$$

$$= \left|\sum_{i=1}^{d} \sum_{j=1}^{d} \sum_{a=1}^{K} y(i) \cdot X_{a,i} \cdot \pi_\theta(a) \cdot (r(a) - \pi_\theta^\top r) \cdot X_{a,j} \cdot y(j)\right. \tag{148}$$

$$- \sum_{i=1}^{d} \sum_{j=1}^{d} \sum_{a=1}^{K} y(i) \cdot X_{a,i} \cdot \pi_\theta(a) \cdot (r(a) - \pi_\theta^\top r) \cdot \sum_{a'=1}^{K} \pi_\theta(a') \cdot X_{a',j} \cdot y(j) \tag{149}$$

$$\left.- \sum_{i=1}^{d} \sum_{j=1}^{d} \sum_{a=1}^{K} y(i) \cdot X_{a,i} \cdot \pi_\theta(a) \cdot \sum_{a'=1}^{K} \pi_\theta(a') \cdot \left(X_{a',j} - \sum_{a''=1}^{K} \pi_\theta(a'') \cdot X_{a'',j}\right) \cdot r(a') \cdot y(j)\right|, \tag{150}$$

which is equal to,

$$|y^\top S y| = \left|\sum_{a=1}^{K} [Xy](a) \cdot \pi_\theta(a) \cdot (r(a) - \pi_\theta^\top r) \cdot [Xy](a)\right. \tag{151}$$

$$- \sum_{a=1}^{K} [Xy](a) \cdot \pi_\theta(a) \cdot (r(a) - \pi_\theta^\top r) \cdot \sum_{a'=1}^{K} \pi_\theta(a') \cdot [Xy](a') \tag{152}$$

$$\left.- \sum_{a=1}^{K} [Xy](a) \cdot \pi_\theta(a) \cdot \sum_{a'=1}^{K} \pi_\theta(a') \cdot r(a') \cdot \left([Xy](a') - \sum_{a''=1}^{K} \pi_\theta(a'') \cdot [Xy](a'')\right)\right|. \tag{153}$$

Denote

$$H(\pi_\theta) := \text{diag}(\pi_\theta) - \pi_\theta \pi_\theta^\top \in \mathbb{R}^{K \times K}. \tag{154}$$

We have,

$$\left|y^\top S y\right| = \left|\left(H(\pi_\theta)\, r\right)^\top (Xy \odot Xy) - \left(H(\pi_\theta)\, r\right)^\top (Xy) \cdot \left(\pi_\theta^\top Xy\right) - \left(\pi_\theta^\top Xy\right) \cdot \left(H(\pi_\theta) Xy\right)^\top r\right| \tag{155}$$

$$= \left|\left(H(\pi_\theta)\, r\right)^\top (Xy \odot Xy) - 2 \cdot \left(H(\pi_\theta)\, r\right)^\top (Xy) \cdot \left(\pi_\theta^\top Xy\right)\right|, \tag{156}$$

where $\odot$ is Hadamard (component-wise) product. According to the triangle inequality and Hölder's inequality, we have,

$$\left|y^\top S y\right| \le \left|\left(H(\pi_\theta)\, r\right)^\top (Xy \odot Xy)\right| + 2 \cdot \left|\left(H(\pi_\theta)\, r\right)^\top (Xy)\right| \cdot \left|\pi_\theta^\top Xy\right| \tag{157}$$

$$\le \|H(\pi_\theta) r\|_\infty \cdot \|Xy \odot Xy\|_1 + 2 \cdot \|H(\pi_\theta) r\|_1 \cdot \|Xy\|_\infty \cdot \|\pi_\theta\|_1 \cdot \|Xy\|_\infty \tag{158}$$

$$= \|H(\pi_\theta) r\|_\infty \cdot \|Xy\|_2^2 + 2 \cdot \|H(\pi_\theta) r\|_1 \cdot \|Xy\|_\infty^2 \qquad \left(\|Xy \odot Xy\|_1 = \|Xy\|_2^2,\ \|\pi_\theta\|_1 = 1\right) \tag{159}$$

$$\le \|H(\pi_\theta) r\|_\infty \cdot \|Xy\|_2^2 + 2 \cdot \|H(\pi_\theta) r\|_1 \cdot \|Xy\|_2^2. \qquad \left(\|Xy\|_\infty \le \|Xy\|_2\right) \tag{160}$$

For $a \in [K]$, denote by $H_{a,:}(\pi_\theta)$ the $a$-th row of $H(\pi_\theta)$ as a row vector. Then,

$$\|H_{a,:}(\pi_\theta)\|_1 = \pi_\theta(a) - \pi_\theta(a)^2 + \pi_\theta(a) \cdot \sum_{a' \neq a} \pi_\theta(a') \tag{161}$$

$$= \pi_\theta(a) - \pi_\theta(a)^2 + \pi_\theta(a) \cdot (1 - \pi_\theta(a)) \tag{162}$$

$$= 2 \cdot \pi_\theta(a) \cdot (1 - \pi_\theta(a)) \tag{163}$$

$$\le \frac{1}{2}. \qquad (\text{using } x \cdot (1-x) \le 1/4 \text{ for all } x \in [0,1]) \tag{164}$$

On the other hand,

$$\|H(\pi_\theta) r\|_1 = \sum_{a \in [K]} \pi_\theta(a) \cdot \left|r(a) - \pi_\theta^\top r\right| \tag{165}$$

$$\le \max_{a \in [K]} \left|r(a) - \pi_\theta^\top r\right| \tag{166}$$

$$\le 2 \cdot \|r\|_\infty. \qquad \left(\text{using } r \in \left[-\|r\|_\infty, \|r\|_\infty\right]^K\right) \tag{167}$$

Therefore, we have,

$$\left|y^\top S(X, r, \theta)\, y\right| \le \|H(\pi_\theta) r\|_\infty \cdot \|Xy\|_2^2 + 2 \cdot \|H(\pi_\theta) r\|_1 \cdot \|Xy\|_2^2 \tag{168}$$

$$= \max_{a \in [K]} \left|\left(H_{a,:}(\pi_\theta)\right)^\top r\right| \cdot \|Xy\|_2^2 + 2 \cdot \|H(\pi_\theta) r\|_1 \cdot \|Xy\|_2^2 \tag{169}$$

$$\le \max_{a \in [K]} \|H_{a,:}(\pi_\theta)\|_1 \cdot \|r\|_\infty \cdot \|Xy\|_2^2 + 4 \cdot \|r\|_\infty \cdot \|Xy\|_2^2 \tag{170}$$

$$\le \left(\frac{1}{2} + 4\right) \cdot \|r\|_\infty \cdot \|Xy\|_2^2 \tag{171}$$

$$\le \frac{9}{2} \cdot \|r\|_\infty \cdot \|X\|_{\mathrm{op}}^2 \cdot \|y\|_2^2 \tag{172}$$

$$= \frac{9}{2} \cdot \|r\|_\infty \cdot \lambda_{\max}(X^\top X) \cdot \|y\|_2^2, \tag{173}$$

where $\|X\|_{\mathrm{op}}$ is the operator norm of $X \in \mathbb{R}^{K \times d}$ (squared root of largest eigenvalue of $X^\top X$),

$$\|X\|_{\mathrm{op}} = \sup\left\{\|Xv\|_2 : \|v\|_2 \le 1,\ v \in \mathbb{R}^d\right\}. \tag{174}$$

According to Taylor's theorem, for all $\theta,\ \theta' \in \mathbb{R}^d$, there exists $\theta_\zeta := \zeta \cdot \theta + (1 - \zeta) \cdot \theta'$ with $\zeta \in [0, 1]$, such that,

$$\left|(\pi_{\theta'} - \pi_\theta)^\top r - \left\langle \frac{d\pi_\theta^\top r}{d\theta}, \theta' - \theta \right\rangle\right| = \frac{1}{2} \cdot \left|(\theta' - \theta)^\top S(X, r, \theta_\zeta)\, (\theta' - \theta)\right| \tag{175}$$

$$\le \frac{9}{4} \cdot \|r\|_\infty \cdot \lambda_{\max}(X^\top X) \cdot \|\theta' - \theta\|_2^2. \qquad \square$$

**Lemma 4** (Alternative expression of co-variance). *Given any vectors $x \in \mathbb{R}^K$, $y \in \mathbb{R}^K$, we have, for all policy $\pi \in \Delta(K)$,*

$$\text{Cov}_\pi(x, y) = \sum_{i=1}^{K-1} \pi(i) \cdot \sum_{j=i+1}^{K} \pi(j) \cdot (x(i) - x(j)) \cdot (y(i) - y(j)). \tag{176}$$

*Proof.* Note that, $\text{Cov}_\pi(x, y) = x^\top \left(\text{diag}(\pi) - \pi\pi^\top\right) y$. Next, we have,

$$x^\top \left(\text{diag}(\pi) - \pi\pi^\top\right) y = \sum_{i=1}^{K} \pi(i) \cdot x(i) \cdot y(i) - \sum_{i=1}^{K} \pi(i) \cdot y(i) \cdot \sum_{j=1}^{K} \pi(j) \cdot x(j) \tag{177}$$

$$= \sum_{i=1}^{K} \pi(i) \cdot x(i) \cdot y(i) - \sum_{i=1}^{K} \pi(i)^2 \cdot x(i) \cdot y(i) - \sum_{i=1}^{K} \pi(i) \cdot y(i) \cdot \sum_{j \neq i} \pi(j) \cdot x(j) \tag{178}$$

$$= \sum_{i=1}^{K} \pi(i) \cdot x(i) \cdot y(i) \cdot (1 - \pi(i)) - \sum_{i=1}^{K} \pi(i) \cdot y(i) \cdot \sum_{j \neq i} \pi(j) \cdot x(j) \tag{179}$$

$$= \sum_{i=1}^{K} \pi(i) \cdot x(i) \cdot y(i) \cdot \sum_{j \neq i} \pi(j) - \sum_{i=1}^{K} \pi(i) \cdot y(i) \cdot \sum_{j \neq i} \pi(j) \cdot x(j) \tag{180}$$

$$= \sum_{i=1}^{K-1} \pi(i) \cdot \sum_{j=i+1}^{K} \pi(j) \cdot (x(i) \cdot y(i) + x(j) \cdot y(j)) - \sum_{i=1}^{K-1} \pi(i) \cdot \sum_{j=i+1}^{K} \pi(j) \cdot (x(j) \cdot y(i) + x(i) \cdot y(j)) \tag{181}$$

$$= \sum_{i=1}^{K-1} \pi(i) \cdot \sum_{j=i+1}^{K} \pi(j) \cdot (x(i) - x(j)) \cdot (y(i) - y(j)), \tag{182}$$

finishing the proofs. $\qquad\square$

## C  Generalization to MDPs

We discuss some research plans for generalizing the results to MDPs, considering Softmax PG for illustration. The discussion provides some new ideas, but resolving this problem is highly non-trivial and requires further investigation. We omit the introduction of notations for general finite MDPs.

According to the policy gradient theorem [25, Theorem 1], we have, for all $\theta \in \mathbb{R}^d$,

$$\theta_{t+1} = \theta_t + \eta \cdot \sum_{s \in \mathcal{S}} d^{\pi_{\theta_t}}(s) \cdot \sum_{a \in \mathcal{A}} \frac{\partial \pi_{\theta_t}(a|s)}{\partial \theta_t} \cdot Q^{\pi_{\theta_t}}(s, a) \tag{183}$$

$$= \theta_t + \eta \cdot \sum_{s \in \mathcal{S}} d^{\pi_{\theta_t}}(s) \cdot X_s^\top \left(\text{diag}(\pi_{\theta_t}(\cdot|s)) - \pi_{\theta_t}(\cdot|s)\pi_{\theta_t}(\cdot|s)^\top\right) Q^{\pi_{\theta_t}}(s, \cdot), \tag{184}$$

where $X_s \in \mathbb{R}^{|\mathcal{A}| \times d}$ is the feature matrix under state $s \in \mathcal{S}$ and can be shared across multiple states. Comparing with Eq. (4), for all $s \in \mathcal{S}$, the reward vector $r \in \mathbb{R}^K$ is replaced with $Q^{\pi_{\theta_t}}(s, \cdot)\mathbb{R}^{|\mathcal{A}|}$, which provides some new ideas as well as difficulties.

The idea is that preserving the order of $Q^*(s, \cdot)$ (value of the optimal policy $\pi^*$ under state $s \in \mathcal{S}$) might be enough to achieve global convergence. Here we show a local convergence when $\text{softmax}(X_s\theta_t)$ is close enough to $\pi^*(\cdot|s)$. Suppose that there exists $w \in \mathbb{R}^d$, such that for all $s \in \mathcal{S}$, $X_s w \in \mathbb{R}^{|\mathcal{A}|}$ preserves the order of $Q^*(s, \cdot)$. For any $\theta_t$ such that $Q^{\pi_{\theta_t}}(s, \cdot)$ preserves the order of $Q^*(s, \cdot)$, we have,

$$\theta_{t+1}^\top w = \theta_t^\top w + \eta \cdot \sum_{s \in \mathcal{S}} d^{\pi_{\theta_t}}(s) \cdot w^\top X_s^\top \left(\text{diag}(\pi_{\theta_t}(\cdot|s)) - \pi_{\theta_t}(\cdot|s)\pi_{\theta_t}(\cdot|s)^\top\right) Q^{\pi_{\theta_t}}(s, \cdot) \tag{185}$$

$$\geq \theta_t^\top w, \tag{186}$$

which is similar to and generalizes Eq. (12). If one can show that $\theta_t$ approaches $w$ in direction, then $\pi_{\theta_t}(a^*(s)|s) = \text{softmax}(X_s\theta_t)(a^*(s)|s) \to \pi^*(a^*(s)|s) = 1$. This means that preserving the order of $Q^*(s, \cdot)$ could be enough for $\pi^*$ to be a local attractor for Softmax PG updates. One challenge is to generalize the arguments for arbitrary initialization $\theta_1 \in \mathbb{R}^d$ rather than $\theta_t$ being close enough

to optimal solution, and the difficulty is that $Q^{\pi_{\theta_t}}(s, \cdot)$ does not necessarily preserve the order of $Q^*(s, \cdot)$, and the above inequality does not necessarily hold.