# OpenReview forum: "Ordering-based Conditions for Global Convergence of Policy Gradient Methods"
_NeurIPS.cc/2023/Conference — NeurIPS 2023 oral_

### Official Review · Reviewer_BpdK · 2023-07-01

**Soundness:** 3 good
**Presentation:** 4 excellent
**Contribution:** 4 excellent
**Rating:** 8
**Confidence:** 3

**Summary:**

The paper theoretically proves that, for finite-arm bandits with linear function approximation, the global convergence of policy gradient (PG) methods is not dependent on approximation error, but rather, on the ordering properties of the reward representation. The global convergence is achievable for both standard Softmax PG and natural policy gradient (NPG) under linear function approximation. This result is also verified using simple examples and empirical experiments.

**Strengths:**

This paper provides a completely new understanding on the convergence of policy gradient method, that is, the global convergence only depends on the specific ordering-based condtions instead of some classical approximation requirements. To my own knowledge, I have not seen such result before.

This novel insight is also significant. There are many situations where the approximation error could never be sufficiently small. This work provides a key to understand the covnergence of PG-based algorithms. And as the author claims, I agree with that this paper will open a new directions for PG-based methods under function approximation.

This work is well-written. I can clearly understand which problem this paper is solving; especially, examples given in Section 3 are very helpful to understand the motivation of proposing the ordering-based conditions. This paper also has rigorously defined two new preservation condtions on the ordering, and proved the necessarity or sufficiency for those proposed conditions.

**Weaknesses:**

The studied case only contains one state, so it is a simple bandit optimization problem. Though the main result of this paper is inspiring and interesting, this paper lacks of (1) persuading evidence showing that these understandings will also be valid for more general cases, and (2) sufficient discussions on how to generalize this result in a more general Markov decision process.


**Questions:**

The single-state MDP is a very special MDP; the proposed idea may or may not work for a more general senario. How this work will be generalized to a multi-state MDP? Is there any simulation results on that?

**Limitations:**

This is a theoretical work so there is no negative societal impact.

---

> ### Author Rebuttal · Authors · 2023-08-10
>
> We appreciate that the reviewer recognized the contribution of the work. We answer the questions as follows.
>
> >**discussion of how to generalize the results to MDPs**
>
> Generalizing the results to MDPs is an important and challenging next step, as we mentioned in the conclusions. Since other reviewers (P4aQ, m2vF) also asked this question, we present the discussions for MDPs in the common feedback, and please check that for details. Thank you.

---

> > ### Comment · Reviewer_BpdK · 2023-08-14
> > **Response**
> >
> > Thank you for the detailed explanation regarding the generalization to MDPs. It further confirms my belief that this paper has made a significant contribution and could potentially lead the way for a new direction. I have read comments from other reviewers, and none raised any new concerns for me. So I will keep my current positive scores.

---

### Official Review · Reviewer_WYfH · 2023-07-06

**Soundness:** 3 good
**Presentation:** 4 excellent
**Contribution:** 4 excellent
**Rating:** 8
**Confidence:** 5

**Summary:**

This work studies the global convergence of standard softmax policy gradient and natural policy gradient for finite-arm bandits with linear function approximation (i.e., considering a log-linear policy). It is shown that the approximation error is not crucial for characterizing global convergence since the latter can be achieved even with non-zero approximation error. Ordering-based conditions are provided instead to guarantee the global convergence of softmax PG and natural PG while the approximation error is non-zero. More precisely, this paper establishes that both NPG and softmax PG converge globally when (a)  for NPG, the optimal action’s rank is preserved by the projection of the true reward onto the space of representable rewards and (b) for softmax PG, there exists a linear function preserving the ranking of actions provided by the true reward function. The case of a linearly realizable reward function is a particular case of the aforementioned reward rank preservation condition. Numerical examples illustrate all these results throughout the paper.


**Strengths:**


The main results of this paper provide very interesting insights on the global convergence of some PG methods under the function approximation setting for finite-arm bandits. The paper provides strong and solid contributions.

**(a) Originality**: the paper challenges common existing analysis featuring the approximation error as a natural limit when considering function approximation. Results are new to the best of my knowledge.

**(b) Significance**: as also highlighted by the paper in the conclusion, the results are likely to inspire further important developments for function approximation in the more general setting of MDPs, nonlinear approximation and representation learning.

**(c) Correctness**: To the best of my knowledge, except for the proof of Theorem 1 which is more involved and may deserve some clarifications in my opinion (see my comments below), the proofs in the appendix are correct, complete and relatively easy to follow. I checked the appendix in details.

**(d) Clarity and writing**: This paper is very well-written, the structure is clear, the exposition is progressive, the story line is very nice and the illustrations are insightful. This is a solid body of work.


**Weaknesses:**

- While the proof of Theorem 2 is very clear to me, some parts of the proof of Theorem 1 (and its sketch l. 266 to 293) deserve some clarifications in my opinion. Please see my comments in the questions section.

- Comment: The proof of Theorem 2 seems to follow similar lines to the proofs in Khodadadian et al. 2022 even if the latter work does not consider linear function approximation nor does it propose ordering-based conditions (while it applies however to MDPs).  The main difference seems in replacing the  true gap $\Delta$ by the gap induced by the linear approximation $\hat{\Delta}$. For similarities, see for e.g., Appendix C in Khodadadian et al. 2022 for the bandits case or the main part of the paper (Theorem 1, Lemma 1, Proposition 1 p. 3-4 and their proofs).

Khodadadian, S., Jhunjhunwala, P. R., Varma, S. M., & Maguluri, S. T. (2022). On linear and super-linear convergence of Natural Policy Gradient algorithm. Systems & Control Letters, 164, 105214.

**Minor:**

-  It is not very clear what is $v_2$ in l. 278-279 (especially that there is also $v$ in l. 278) and what is $v_1$ in l. 283 and how do they relate to $v$ in l. 268, I find the notations and the formulation a bit confusing in l. 266-293.

- Please give the precise reference to Theorem 1 when you reference [8] in lines 116 and 123.

- $r(a^*)$ introduced in l. 110 does not seem to be defined before (definition of $a^*$ comes later in l. 140).

- Numerical legends in all the figures are almost invisible in a printed format (although visible when zooming on a computer), it would be nice to increase the size of the numerical characters (and save the figure in pdf format if not already done) to ease the reading.

- Eq. (27): Why is the inequality strict? This (strict) positivity does not seem to be used anyway.

- Suggestion: Eq. (102) maybe add $>0$ for clarification.

- Suggestion: Eq. (149) to Eq. (150) maybe add the splitting sum steps with the order summing exchange to detail a bit more for the convenience of the reader.

**Typos:**
- l. 446: $\theta’ \in \mathbb{R}^d$ instead of $\mathbb{R}^K$?

- l. 474: $a \in [K]$ instead of $i \in [a]$.

- l. 481: Eq. (58) instead of (54).

- l. 493: Eq. (68) instead of (65).

- Eq. (97): Eq. (94) instead of (91).

- Eq. (101): Eq. (97) instead of (95).

- Eqs. (138) and (139): $\max_a$ instead of $\max_i$.

- l. 539, Eq. (144): say there exists $\theta_{\zeta} \in [\theta, \theta']$ such that ...




**Questions:**

About the proof of Theorem 1, I have the following technical questions (1. (a)-(b)) which constitute my main concerns and that I would like to be addressed for clarification before updating my evaluation:

**1. (a)** Second part/Lemma 1 (l. 455): Why does the fact $\|\| \frac{d \pi_{\theta_t}^T r}{d \theta_t}\| \|_2 \to 0$ imply that $\|\|\theta_t\|\| \to \infty$? I understand that there is no stationary point for any $\theta \in \mathbb{R}^d$ following the proof but how do you conclude properly that then $\|\|\theta_t\|\| \to \infty$? In particular in the proof, l. 446: what does the assumption
suppose there exists $\theta' \in \mathbb{R}^K$ with $\|\|\theta’\|\|_2 < \infty$ mean?

It is proven (by contradiction) that for every $\theta’ \in \mathbb{R}^d$, we have $\frac{d \pi_{\theta’}^T r}{d \theta’} \neq 0$.
To conclude that $\|\|\theta_t\|\| \to \infty$, I would expect that the proof assumes that the sequence $(\theta_t)$ is bounded and then finds a contradiction. Could you please clarify the reasoning here?

**1. (b)** Third part: the general reasoning and the conclusion l. 498 to 501 are not very clear to me.

   (i) It is proven by contradiction that every suboptimal action $i$ s.t. (47) is satisfied, $\pi_{\theta_t}(i)$ does not converge to 1. How does this help concluding that for the optimal action $a$ (or any optimal action if non unique) we have $\pi_{\theta_t}(a)$ converges to 1? Could you clarify the reasoning and the guiding intuition?

 (ii) Could you also elaborate more on how (60), (69), (71) and (72) are used to conclude?

(iii) Is the third case $j \in \mathcal{A}(i)$ (i.e., $i,j$ are mapped to the same reward value) handled in the proof? If trivial, maybe a word about it can be added.

 (iv) Minor: what do you mean by $X$ and $u$ are bounded in l. 492? $X$ is the feature matrix and $u$ is a fixed vector in $\mathbb{R}^d$.

**2.** Could you clarify why the possibility of having exponentially many suboptimal local maxima renders global convergence (to the optimal reward) impossible without further structure on function approximation as stated in l. 123-124? Is it because there is more chance to be stuck at a local suboptimal local maxima? How does the ordering-based condition preclude such a case in Theorem 1?

**3.** Do the examples of section 3 allude to the fact that the ordering of the features matters since the examples only differ between them by column permutations?

**4.** In proposition 1 (and more generally in the paper), do you assume that there exists a unique optimal action? If there are many, I guess any optimal action would work. You may want to add a comment regarding this.


**Limitations:**

Limitations and opportunities for future work are briefly discussed in the conclusion.

---

> ### Author Rebuttal · Authors · 2023-08-10
>
> Thank you for carefully reading and checking the results. The main concerns are addressed as follows.
>
> >**Comment on Khodadadian et al. 2022**
>
> Thank you for pointing this out. We will add a remark to mention the similarities and differences between Theorem 2 and Khodadadian et al. 2022.
>
> >**1. (a)**
>
> Suppose there exists $C < \infty$, s.t. for all $t \ge 1$, $\theta_t \in S_C := \\{ \theta \in \mathbb{R}^d: \\| \theta \\|_2 \le C \\}$.
>
> For all $\theta \in S_C$, $\Big\\| \frac{d \pi\_{\theta}^\top r}{d \theta} \Big\\|_2 > 0$ (no finite stationary points). Since $S_C$ is compact, we have, $\inf\_{\theta \in S_C}{ \Big\\| \frac{d \pi\_{\theta}^\top r}{d \theta} \Big\\|_2 } \ge \epsilon$ for some $\epsilon > 0$.
>
> Therefore, for all $t \ge 1$, $\Big\\| \frac{d \pi\_{\theta_t}^\top r}{d \theta_t} \Big\\|_2 \ge \epsilon > 0$, contradicting Eq. (28).
>
> >**1. (b)**
>
> (i) The whole $\theta_t \in \mathbb{R}^d$ space is partitioned into at most $K$ sub-regions (Figure 2). Since the existence of $w \in \mathbb{R}^d$, $a^*$ has one sub-region (yellow in Figure 2) which satisfies (by order-preservation),
> \begin{equation*}
>     \[ X \theta_t \](a^*) = \max\_{a \in [K]} \[ X \theta_t \](a)
> \end{equation*}
>
> Since $\\| \theta_t \\|_2 \to \infty$, if **$\theta_t$ always stays in the above region as $t \to \infty$**, then softmax transform gives $\pi\_{\theta_t}(a^*) \to 1$. We proved that for all $i \in [K]$ with $r(i) < r(a^*)$, $\theta_t$ cannot stay in sub-optimal regions forever.
>
> In addition, $\theta_t$ cannot switch between different regions. As $\\| \theta \\|_2 \to \infty$, $\pi\_{\theta}^\top r \approx r(i)$ in a region, and switching regions contradicts $\pi\_{\theta\_{t+1}}^\top r \ge \pi\_{\theta_t}^\top r $ in Eq. (10). Moreover, $\theta_t$ cannot approach region boundaries, since it makes $\pi\_{\theta_t}$ approach non one-hot policies, such as $(0, 0.5, 0.5, 0)$ in Figure 2, which gives non-zero inner product in Eq. (33), contradicting Eq. (28).
>
> **In summary**, $\\| \theta \\|_2 \to \infty$ and $\theta_t$ can only stay in one region, and we proved that $\theta_t$ cannot stay in any sub-optimal regions, which implies that $\theta_t$ eventually stays in the optimal region, indicating that $\pi\_{\theta_t}(a^*) \to 1$.
>
> (ii) Eqs. (69) and (71) give $w^\top \theta_t  \gg 0 \ge u^\top \theta_t$ (the former is unbounded and the later is decreasing), which holds if scaling $w$ and $u$ by constants to get $w'$ and $u'$. For a large enough $t \ge 1$, there exists $w'$ such that $x_{a^*}^\top \theta_t \ge {w'}^\top \theta_t > {u'}^\top \theta_t \ge x_{a^-}^\top \theta_t$ (geometry argument, $u$ can be arbitrarily close to region boundaries). The direction of $\theta_t$ approaches $v$ implies $x_{a^*}^\top v > x_{a^-}^\top v$, meaning that $\theta_t$ enters the region in Case 1, contradicting the assumption of Case 2.
>
> For example, take $u \propto (-0.9, 1)^\top$ and $w \propto (-0.9, -1)^\top$ in Example 1 (Figure 2), $w^\top \theta_t > u^\top \theta_t$ implies $\theta_t(2) < 0$, entering the dark green region (reducing to Case 1). Another example is if $\\| x_a \\|_2 = 1$ for all $a$ (features are on unit ball), then $u' = x\_{a^-}$ satisfies the inequalities.
>
> (iii) If $j \in \mathcal{A}(i)$, then consider the third largest component and discuss the two cases again. If $j \in \mathcal{A}(i)$ for all components, then $\pi_{\theta_t}$ approaches uniform policy $\text{softmax}(X v) = \text{softmax}(\mathbf{1})$, contradicting $\theta = \mathbf{1}$ cannot be a stationary point (unless $ r = \mathbf{1}$ according to Eq. (33), which is trivial since every policy is optimal).
>
> (iv) This means there exists $C < \infty$, such that $\max_{i \in [K], j \in [d]} | X_{i,j} | \le C$ and $\max_{j \in [d]} | u(j) | < C$. The result of this is for $y$ in Line 492, $\max_{i \in [K]} | y(i) | $ is also bounded since $y := X u$ in Line 489.
>
> >**2.**
>
> In [1, Thm 1], if $\theta_1$ is close enough (in a basin of attraction) to one bad local maximum, then gradient ascent will make $\theta_t$ approach that bad local maximum.
>
> The ordering-based condition preclude such a case since **first**, Lemma 1 shows that there is no stationary point in any finite region. **Second**, the landscape is "ordered" such that any sub-optimal one-hot policy is not a bad local maximum (rigorously speaking they are not even stationary points by being infinitely far). They are "saddle-point-like", in the sense of being surrounded by a higher and a lower plateau. This creates a situation that any sub-optimal one-hot is not attracting gradient trajectories, and the optimal plateau is the only "local-maximum-like stationary point" in Figure 1(a).
>
> >**3.**
>
> This is partially correct. If we change the numbers in Example 1 and use the same Examples 2 and 3, the same conclusion still holds, e.g., use $r = (9.2, 7.8, 7.1, 6.3)^\top$ and $X^\top = \[ -0.2, -1.1, 0.1, 2.1 ; -1.9, 0.2, 0.8, 0.1 \]$ as Example 1.
>
> >**4.**
>
> Having multiple optimal actions does not change the conclusion of $\pi_{\theta_t}^\top r \to r(a^*)$. The difference is that the behaviour of optimal actions' probabilities needs to be discussed in a case by case manner, depending on whether different optimal actions' plateaus are connected or separated on landscape.
>
> Consider if $a_1^*$ and $a_2^*$ with $r(a_1^*) = r(a_2^*)$. If the plateaus for those two actions are separated on the landscape in Figure 1 (this is an intuitive statement), then we will have either $\pi_{\theta_t}(a_1^*) \to 1$ or $\pi_{\theta_t}(a_2^*) \to 1$ depending on where the initialization $\theta_1$. Otherwise,  If the plateaus for those two actions are connected on the landscape (neighborhoods of one another), then we will have $\pi_{\theta_t}(a_1^*) + \pi_{\theta_t}(a_2^*) \to 1$, since the current analysis only gives $\theta_t$ approach the "joint" optimal plateau formed by $a_1^*$ and $a_2^*$.
>
> >**Typos and Minor comments**
>
> Thank you for catching typos and minor issues.  We will ensure these are fixed.

---

> > ### Comment · Reviewer_WYfH · 2023-08-14
> > **Post rebuttal**
> >
> > I thank the authors for their precise and satisfactory answers to my questions and concerns. I believe this is an interesting and solid paper which provides nice and novel insights about the convergence of PG methods in the linear function approximation setting and which has some potential to be extended to the more challenging MDP setting (despite  important obstacles), the discussion provided by the authors as a general comment (in the rebuttal) regarding this last point is insightful. I raised my score to 8. Here are some minor additional follow-up comments below.
> >
> > - I read the other reviews and the authors’ responses and I think the discussion regarding the verification of ordering-based conditions following the other reviewers’ comments could be added to the paper.
> >
> > - 1.b I think the proof in the appendix can be revised to reflect the intuition provided in the rebuttal as a response to 1. b. (i) in particular. I find the overall writing of the third part of the proof rather confusing although the treatment of each of the subcases of the partition is rather clear. I refer in particular to the beginning: Why considering the case where the limit of normalized $\theta_t$ sequence converges to a constant vector in (47)? It is not very clearly discussed in the appendix what would happen if the normalized sequence does not converge to a fixed vector but rather rotates indefinitely (spiralling towards $+\infty$). This point was clearer to me after the rebuttal given the responses to 1. b (i) and (ii).
> >
> > - 3. A comment added to the paper regarding this would probably be useful, I guess ordering of the features should not be crucial in examples since feature matrices would induce the same linear space. I think it is relevant to show that examples are not only based on column permutations.
> >
> > - 4. Thank you for the clarification, please update the paper accordingly to clarify this case where there are multiple optimal actions since it is not commented on this.

---

### Official Review · Reviewer_P4aQ · 2023-07-18

**Soundness:** 4 excellent
**Presentation:** 4 excellent
**Contribution:** 4 excellent
**Rating:** 9
**Confidence:** 4

**Summary:**

This paper challenges (arguably) the current best known policy gradient (PG) convergence analysis, which is the conventional approximation error based analysis originally proposed by the seminal work of Agarwal et al. (2021). To this end, the authors consider the finite-arm bandits with log-linear policy and study the conditions of the global convergence of PG and natural policy gradient (NPG).

**First**, by carefully designing numerical simulations, the authors show that global convergence can be achieved even if the parameterized policy space can not cover the full policy space, or the approximation error is not zero. Consequently, the approximation error is not a key quantity for characterizing global convergence in either algorithm under linear function approximation.

**Second**, the authors establish new conditions of the global convergence of PG and NPG for the same setting, separately. For NPG, the necessary and sufficient condition of the global convergence is whether the projection of the reward vector onto the feature map strictly preserves the top ranking of the optimal action. For PG, the sufficient but not necessary condition of the global convergence is whether there exists a point in the image of the feature map such that it preserves the entire ranking of the reward vector. These conditions are again well supported by numerical simulations.



Agarwal, Alekh, Sham M. Kakade, Jason D. Lee, and Gaurav Mahajan (2021). On the Theory of Policy Gradient Methods: Optimality, Approximation, and Distribution Shift. Journal of Machine Learning Research 22.98, pp. 1–76.

**Strengths:**

The paper is revolutionary and the results are surprising. This paper may have the same impact on the theoretical RL community as that of Agarwal et al. (2021). The strengths of the paper can be summarized as follows.

- First, the research question itself on the approximation error assumption is important and revolutionary. Challenging the previous pioneering work is always not easy.
- However, the argumentation of the paper is impeccable. Readers will probably be surprised at first, but will quickly be convinced by several simple but sophisticated numerical simulations.
- Not only the authors are able to find negative answers to questions about the role of approximation error for the global convergence of PG methods, they also establish new results characterizing the conditions for the global convergence of PG methods and draw the connection between the new conditions and the approximation error assumption. The novelty of the paper is significant.
- In addition, the paper is very well written. The research question is well formulated. The new conditions are well presented. And the reasoning is detailed with intuitive explanations, figures, many examples and the proof sketch.

I agree with the authors that this work will open many new directions for understanding PG-based methods in the function approximation regime, especially considering the general Markov decision processes (MDPs).

**Weaknesses:**

Although the paper is well written, I still find some minor points for improvement.

- In the figures, the authors can specify that y-axis is the reward / value function.
- Line 52-53: "... approximation error ..., diverting attention from feature designs that achieve useful properties beyond small approximation error." Although one goal of the paper is to claim that approximation error is not necessary, I find this sentence to be a little too dismissive of approximation error. It may leave the reader with the impression that approximation error is rarely useful outside of the tabular case where the approximation error is zero. It is true that, zero approximation error does not fit well with linear function approximation in general, which is the original motivation for the paper. However, when the approximation error is zero, the conventional approximation error based analysis becomes useful. For instance, another interesting case of zero approximation error is the use of neural networks, recently studied by Alfano et al. (2023) in Section 4.2. That is, a sufficiently wide and shallow ReLU network can infinitely approximate the Q-function such that the approximation error is zero. Consequently, their approximation error based analysis leads to a new SOTA sample complexity of PG method under neural network parameterization.
- Line 361-362: "Extending the results and techniques to general Markov decision processes (MDPs) is another important and challenging next step." The authors can use an extra page in the revised version to discuss the intuition of possible obstacles to such an extension. See also my question below.

Alfano, Carlo, Rui Yuan, and Patrick Rebeschini (2023). A Novel Framework for Policy Mirror Descent with General Parametrization and Linear Convergence.

**Questions:**

First, I have a clarification question. The authors claim that they solve an open problem left by Agarwal et al. (2021). If I understand correctly, the authors refer to this one: _if zero approximation error holds, does softmax PG achieve global convergence ?_ However, I couldn't find this claim in Agarwal et al. (2021). Instead, I found two open problems left by Agarwal et al. (2021).

- __(1)__ In their Remark 11 (journal version) / Remark 5.1 (arxiv version), the first open problem is that, whether or not softmax PG will converge globally if the initial state distribution is not state-wise strictly positive. Since this paper only considers bandit problems with one single state, the initial state distribution is constant 1 with that state. The paper does not resolve this open problem.
- __(2)__ In their Remark 14 (journal version / Remark 5.2 (arxiv version), the second open problem is whether a polynomial global convergence rate is achievable for softmax PG with the entropy regularizer. This problem envolves entropy regularization, which is outside the scope of the paper. Thus, the paper dose not resolve this problem neither.

Can the authors point me specifically to where Agarwal et al. (2021) claim  the open problem mentioned right after Corollary 1 in this paper ?

Next, I have some more open questions that the authors can decide to include them in the paper or not.

- Do the results of softmax PG and NPG in this paper still hold if the optimal action is not unique ?
- Same question for entropy regularized bandit case ?
- Do the results for NPG with non-adaptive geometrically increasing step size still hold ? Note that NPG and variants with non-adaptive geometrically increasing step size have been studied extensively recently (Lan, 2022; Xiao, 2022; Li et al., 2022; Yuan et al., 2023; Alfano et al., 2023). Can we obtain superlinear convergence as discussed in Xiao (Section 4.3, 2022) and Li et al. (2022) ?
- As for the general MDP, one can think about the compatible function approximation framework of Agarwal et al. (2021). By similarity, the ranking condition of the reward vector for the softmax PG becomes the ranking condition of the advantage function / Q-function. And the projection of the reward vector onto the image of the feature map for NPG becomes the projection of the advantage function / Q-function onto the image of the matrix $[\nabla_\theta \log \pi_{\theta}(a \mid s)]_{s, a}$. What do you think about this idea ? Can this idea and the techniques used in this paper be enough to be extended to general MDP ? If not, what is the missing factor here and what is the challenging obstacle ? For simplicity, one can consider the exact NPG instead of stochastic NPG.

Xiao, Lin (2022). On the Convergence Rates of Policy Gradient Methods. Journal of Machine Learning Research 23.282, pp. 1–36.

Li, Yan, Tuo Zhao, and Guanghui Lan (2022). Homotopic Policy Mirror Descent: Policy Convergence, Implicit Regularization, and Improved Sample Complexity.

Lan, Guanghui (Apr. 2022). Policy mirror descent for reinforcement learning: linear convergence, new sampling complexity, and generalized problem classes. Mathematical Programming.

Yuan, Rui, Simon Shaolei Du, Robert M. Gower, Alessandro Lazaric, and Lin Xiao (2023). Linear Convergence of Natural Policy Gradient Methods with Log-Linear Policies. In International Conference on Learning Representations.

Alfano, Carlo, Rui Yuan, and Patrick Rebeschini (2023). A Novel Framework for Policy Mirror Descent with General Parametrization and Linear Convergence.

**Limitations:**

The authors have adequately addressed the limitations.

---

> ### Author Rebuttal · Authors · 2023-08-10
>
> We appreciate that the reviewer understood and recognized the contribution of the work. We answer the questions as follows.
>
> >**figures ... y-axis**
>
> We mentioned that $\pi_{\theta}^\top r$ is used as the vertical axis value in Line 148, and will add this to the figure.
>
> >**Line 52-53 ... approximation error ... Alfano et al. (2023)**
>
> Thank you for pointing out the work on PG using overparameterized NNs. We will cite it and mention that zero function approximation error results are useful in this setting, as suggested.
>
> >**Line 361-362**
>
> Thank you for the suggestions. We present our current understanding and evidences. Also, please check the common feedback for details.
>
> >**Clarification question ... open problem**
>
> Thank you for checking Agarwal et al. (2021) very carefully. It is not explicitly mentioned in that paper **if Softmax PG achieves global convergence with zero approximation error** is an open problem. However, to the best of our knowledge and after extensive communication with the authors of Agarwal et al. (2021), the common understanding is that this problem has remained open in the sense that it has not been solved before. We will clarify in revised versions.
>
> >**optimal action is not unique**
>
> Yes, $\pi_{\theta_t}^\top r \to r(a^*)$ will hold. Consider if there are two optimal actions $a_1^*$ and $a_2^*$ with $r(a_1^*) = r(a_2^*)$. If the plateaus for those two actions are separated on the landscape in Figure 1 (this is an intuitive statement), it will follow that either $\pi_{\theta_t}(a_1^*) \to 1$ or $\pi_{\theta_t}(a_2^*) \to 1$ depending on where $\theta_1$ is initialized (the arguments are almost identical as the unique optimal action case). Otherwise,  if the plateaus for those two actions are connected on the landscape (neighborhoods of one another), then we will have $\pi_{\theta_t}(a_1^*) + \pi_{\theta_t}(a_2^*) \to 1$, since the current analysis only asserts that $\theta_t$ approaches the ``joint'' optimal plateau formed by $a_1^*$ and $a_2^*$.
>
> >**entropy regularized bandit case**
>
> Thank you for asking this question. Our speculation is that the answer is yes. In particular, we believe that when the temperature is small enough (entropy does not have much weight comparing to reward), the landscape is modified so that the stationary point is "moved" from a one-hot policy to a finite stationary point, hence the nice "ordered" landscape is still preserved by adding entropy. Further study is required to rigorously show or disprove this speculation.
>
> >**NPG with non-adaptive geometrically increasing step size still hold**
>
> Thank you for pointing this out. We have checked and do not see any difficulty that prevents this generalization.
>
> >**As for the general MDP ... advantage function / Q-function ... challenging obstacle**
>
> Thank you for bringing up this idea, which seems to be very reasonable. As mentioned in the common feedback, considering the advantage function / Q-function is also the idea we were looking at. Difficulties are also explained in the common feedback.  We believe additional efforts are needed to make this idea work in MDPs.

---

> > ### Comment · Reviewer_P4aQ · 2023-08-10
> >
> > I thank the authors for their thorough and clear response, especially for the discussion of generalization to MDPs. I encourage the authors to present discussions of other cases in the revised version, such as the case where the optimal action is not unique, mentioned by Reviewer WYfH and myself, the case of entropy regularized bandits, and the linear feasibility problem mentioned by Reviewer hbxK. Overall, I am inclined to maintain my score. I will also pay attention to Reviewer WYfH's upcoming comments to see if their concerns on the proofs are cleared up.

---

### Official Review · Reviewer_m2vF · 2023-07-23

**Soundness:** 2 fair
**Presentation:** 2 fair
**Contribution:** 3 good
**Rating:** 5
**Confidence:** 4

**Summary:**

The paper studies softmax policy gradient and natural policy gradient methods for multi-arm bandits problems using linear function approximation. The authors provide examples to illustrate the global convergence of these methods when the standard function approximation error is not zero. To better characterize the global convergence, the authors provide the ordering-based conditions on rewards. Some numerical examples are provided to verify the proposed conditions.

**Strengths:**

**orginality**

- The studied deterministic policy gradient methods are well-known in the literature. The authors re-revisit the convergence of these methods for multi-arm bandits with linear function approximation. Such convergence hasn't been studied directly.

- The authors show necessary (and sufficient) conditions for two policy gradient methods to converge in the linear function approximation setting. The analysis generalizes some similar analysis, e.g., reference [18] to linear function approximation. Technical comparisons with the existing analysis are not very clear.

**quality**

- It seems that the global convergence is abused in some way, since the existing convergence studies in the linear function approximation investigate different sub-optimality gaps, e.g., [4] is based on the regret analysis while [24] utilizes the mirror descent analysis.

- It is not clear how to interpret 'Approximation error is not a key quantity for characterizing global convergence in either algorithm'. The reference [24] shows that zero approximation error leads to global convergence. Although this result does not show necessity, approximation error is still an important quantity we use in practice.

-  Compared with approximation error, it is less clear how to check optimal action perseveration condition, especially when the action space is large. So, weaknesses haven't been clearly stated.

**clarity**

- The paper is structured well, but it lacks of technical comparisons with existing results.

- The visualization is not clearly stated, e.g., axes, error bars, speed.

**significance**

- Since the function approximation is widely used in reinforcement learning, this work is important. It provides new understandings of policy gradient methods in bandit cases.

**Weaknesses:**

- Technical comparisons with existing results are not detailed, sometimes vague. For instance, in line 96 why (4) and (5) for bandits can be used as RL methods; in line 102 which work provides $1/\sqrt{t}$ rate; line 121, why 'insufficiency'; line 152, how to check global convergence; line 191, what of algorithms the quantity must depend on? Please state claims with concrete justifications.

- Optimal action perseveration generalizes the analysis in reference [18]. It is similar to the gap of optimal action and second optimal action used in literature, e.g., reference [12] and the paper: Regret Analysis of a Markov Policy Gradient Algorithm for Multiarm Bandits. These quantities seem to be known important for global convergence and the authors generalize them to linear function approximation. Therefore, it is important to clarify the connections and position the work in the literature properly.

- The usefulness of proposed ordering-based conditions is still questionable. First, the generalization to MDPs is not provided. Second, it is not clear how to check such conditions when state/action spaces are large, even infinite.

- The paper focuses on multi-arm bandits and deterministic policy gradient methods, which have a large gap with reinforcement learning. This work seems to be still in progress and significant effort is needed to generalize this work for serious publication.

**Questions:**

Please see questions in Strengths and Weaknesses.

Here are some other questions.

- Why is it fair to compare function approximation error with ordering-based conditions or optimal action gap conditions? The function approximation error is a general quantity that does not depend on MDP structures, while ordering-based conditions depend on the MDPs. In a basic tabular case, ordering-based conditions do not necessarily hold, e.g., equal rewards at any action, but the softmax function approximation error is zero. Hence, such conditions do not explain the basic case.

- The convergence rate seems to be vague in the main paper, which has a dependence on the gap between optimal action and second optimal action, which is known in the tabular case [12, 18]. How much does your rate analysis go beyond the existing analysis? any rate improvement?

- For softmax PG, reference [15] shows that it can take exponential time to converge in a hard MDP instance. How does ordering-based condition rule out the hard MDP instance?

- The setup is limited to deterministic algorithms with exact gradients, which is an ideal setting. Can the authors apply them to practical stochastic bandit problems?

**Limitations:**

Yes.

---

> ### Author Rebuttal · Authors · 2023-08-10
>
> The review seems to focus on issues arising from misunderstanding or miscommunication. We hope the following can help clarify matters.
>
> >**global convergence is abused ...**
>
> **First**, global convergence simply means $\pi_{\theta_t}^\top r \to r(a^*)$, i.e., policy's reward approaching that of the optimal policy, as mentioned in Lines 230 and 316. It is hard to imagine a more straightforward definition. **Second**, alternative analyses do not change the meaning of global convergence. Sub-linear regret implies that the averaged performance approaches $r(a^*)$. Our global convergence results are for last-iterate, which implies sub-linear regret.
>
> >**... not clear how to interpret Approximation error ...**
>
> We have explained this clearly in Section 3. **First**, Softmax PG and NPG can still achieve global convergence on Example 1 with non-zero approximation errors. This does not contradict "zero approximation error leads to global convergence". **Second**, Examples 1 and 2 have similar non-zero approximation errors but different algorithm behaviors, which makes it impossible to use non-zero approximation error to discriminate the two examples (global convergence or not). **Finally**, we agree that "approximation error is still an important quantity", but our observations also clearly show that non-zero approximation errors are not enough to characterize global convergence.
>
> >**... less clear how to check optimal action preservation condition**
>
> Checking the condition for NPG is no harder than checking the approximation error. To determine approximation error $\\| \hat{r} - r \\|_2 =  \min\_{w \in \mathbb{R}^d} \\| X w - r \\|_2 $ one needs to calculate projection $\hat{r} := X^\top (X^\top X)^{-1} X^\top r$. Optimal action preservation $\text{argmax}\_{a \in [K]}{\hat{r}(a)} = \text{argmax}\_{a \in [K]}{r(a)}$ can be immediately verified from the same calculation. Checking the condition for PG requires a linear feasibility check (i.e., a special case of an LP), as discussed in the response to Reviewer hbxK. Note that checking conditions both PG and NPG only requires the same problem information ($X$, r, and $\hat{r}$) as checking approximation error.
>
> >**in line 96, (4) and (5) ...**
>
> It is well known that general Softmax PG and NPG methods are foundations of the standard RL methods mentioned. Eqs. (4) and (5) are their updates applied to the one-state setting.
>
> >**in line 102, $1/\sqrt{t}$ ...**
>
> Sorry for the typo. [4, Table 1] contains the correct result of an $O(1/t)$ rate.
>
> >**line 121, insufficiency...**
>
> As Section 3 shows, small approximation error is neither necessary nor sufficient for the global convergence of PG or NPG.
> Zero approximation error (i.e., linear realizability) is not necessary in either case, although it is sufficient for NPG, and proved sufficient for softmax PG for the first time in this paper. We could consider replacing "insufficiency of" with "limitations of".
>
> >**line 152, check global convergence...**
>
> As shown in Figure 3(c), we use sub-optimality gap $(\pi^* - \pi_{\theta_t})^\top r$ to ascertain global convergence.
>
> >**line 191, algorithm dependent ...**
>
> Section 3.3 already demonstrated that the condition must depend on the specific update considered (e.g., Softmax PG vs. NPG). Therefore, one has to study the conditions for Softmax PG and NPG **separately** (rather than one condition for both algorithms).
>
> >**... generalizes the analysis of references [18,12] ...**
>
> Our new results for function approximation are not covered by any of those papers. The gap we consider is for $\hat{r}$, which is different than the reward gap of $r$ in those papers.
>
> >**The usefulness ... large, even infinite.**
>
> **First**, the common author response provides a detailed discussion of the MDP case, illustrating how the ideas provide useful initial insights for MDPs. **Second**, for large action spaces, checking new conditions are no harder than approximation error, as explained above. **Third**, for infinite action spaces, it is also infeasible to exactly determine the approximation error in general.
>
> >**... serious publication.**
>
> Understanding the one-state and deterministic settings are necessary first steps before understanding the more involved MDP and stochastic settings. The significance of our findings seem to be well recognized by the other reviewers.
>
> >**Why is it fair to compare function approximation error ...**
>
> To our knowledge, approximation error (and its variants) has been the only quantity considered for characterizing function approximation quality in PG analysis. It is therefore necessary to discuss approximation error when studying the convergence of PGs with function approximation.
>
> >**approximation error ... does not depend on MDP ...**
>
> The approximation error $\min\_{w \in \mathbb{R}^d}{ \\| X w - r \\|_2}$ clearly depends on the problem quantity $r$, as do the order-based conditions.
>
> >**... do not explain the basic case.**
>
> Consider $r = \mathbf{1} \in \mathbb{R}^K$ as mentioned. Consider $w = \mathbf{0} \in \mathbb{R}^d$, such that $r^\prime = X w = \mathbf{0}$. Note that $r^\prime$ preserves the order of $r$ by definition (for all $i, j \in [K]$, $r(i) > r(j)$ if and only if $r^\prime(i) > r^\prime(j)$ as in Line 229).
>
> >**... any rate improvement?**
>
> Our results address the function approximation case, which is well beyond tabular analyses [12,18]. Improving rates for the tabular setting is not a focus of this work. It is certainly possible to consider increasing stepsizes [23] for improvement, which is beyond the scope of this paper.
>
> >**... hard MDP instance?**
>
> It is not possible to rule out hard MDP instances in general. The tabular case is recovered as a special case (Line 232). If hard MDP instances were avoided, they would be avoided in the tabular case, which would contradict [15].
>
> >**... stochastic bandit problems?**
>
> This is mentioned in the conclusion as future work, and we are working to obtain  results in this direction.

---

> > ### Comment · Reviewer_m2vF · 2023-08-14
> >
> > Thank you for the response. I have improved my score and recommend the authors to take these points in revision.
> >
> > 1. Sub-optimality gaps in different PG methods under linear function approximation are different. The metric used in natural policy gradient (NPG) [4, Table 2] is not the same as $\pi_{\theta_t}^\top r\to r(a^\star)$.
> >
> > 2. The gap between optimal action and second optimal action has been used in literature to show the global convergence of PG methods. The authors shoud discuss and connect this work with them more explicitly.
> >
> > 3. Bandits with function approximation have a large gap with MDPs with function approximation.
> >
> > 4. The ordering-based conditions depend on the order while function approximation error does not. They use different information on MDP.

---

### Official Review · Reviewer_hbxK · 2023-07-25

**Soundness:** 4 excellent
**Presentation:** 3 good
**Contribution:** 4 excellent
**Rating:** 7
**Confidence:** 4

**Summary:**

This paper considered the problem of global convergence condition of policy gradient (PG) methods with linear function approximation motivated by three observations: i) global convergence under linear function approximation can be achieved without policy or reward realizability; 2) approximation error is not a critical factor for global convergence; and 3) conditions for characterizing global conference should be algorithm-dependent. Based on these observations, the authors developed new ordering-based conditions for global convergence of PG methods: i) For Softmax PG, a sufficient condition for global convergence to occur is that the representation preserves the ranking of the rewards; and ii) For natural PG (NPG), the necessary and sufficient condition of global convergence is that the projection of the reward onto the representation space preserves the optimal action’s rank.


**Strengths:**

1. This paper develops a new set of global convergence conditions for PG methods with linear function approximation, which advances the state of the art of understanding PG methods (Softmax PG in particular).

2. The proof strategies and algorithm analysis techniques are novel.

3. Motivating examples in this paper are insightful.


**Weaknesses:**

1. The paper could benefit from constructing a bit more larger-scale experiments.

2. This paper could have some further discussions on the implications of the ordering-based conditions.

**Questions:**

I appreciate the new findings and fresh insights of the ordering-based global convergence conditions for PG methods with linear function approximation. One immediate question that comes to my mind after reading this paper is how restrictive (or non-restrictive) these ordering-based conditions are. For example, for the Softmax PG method, it appears to me that given $r$ and $X$, checking the existence of a $w$ that preserves the reward ranking may not be a simple task. Is there a systematic way to check the existence of such a $w$? Is there any other condition equivalent to order-preserving but easier to identify and check? Could the authors discuss how large the subspace of $X$ in $\mathbb{R}^{d}$ that satisfies the order-preserving condition is? The same questions can also be asked for NPG.

---

> ### Author Rebuttal · Authors · 2023-08-10
>
> We appreciate that the reviewer understands and recognizes the contributions of this work. We address the main concerns as follows.
>
> >**Is there a systematic way to check the existence of such a $w$ given $r$ and $X$:**
>
> Yes, checking the existence of $w$ is known as **linear feasibility** in the literature (Grötschel et al., 2012), i.e., determining whether a set of inequalities has a non-empty intersection. In particular, suppose $X \in \mathbb{R}^{K \times d}$ and $r \in \mathbb{R}^K$ are given and that $r$ is sorted, i.e., $r(1) \ge r(2) \ge \cdots \ge r(K)$. Denote $x_i \in \mathbb{R}^d$ as the $i$th row vector of $X$. The linear feasibility problem in this case is to check if there exists a $w \in \mathbb{R}^d$, such that, for all $i \in [K-1]$,
> \begin{align}
>     x_i^\top w \ge x_{i+1}^\top w.
> \end{align}
> Linear feasibility can be cast as linear programming using a dummy objective and keeping the constraints, hence any LP technique, such as the ellipsoid method, can be used to solve it (Grötschel et al., 2012).
>
> >**How large the subspace of $X$ that satisfies the order-preserving condition is? The question could also be asked for NPG.**
>
>  That is an interesting question. **First**, this work shows that the space is strictly larger than the set of $X$s that satisfy linear realizability / zero approximation error (i.e., the set of $X$ such that there exists $w \in \mathbb{R}^d$ to satisfy $X w = r$). From Line 232 in the paper, we know that zero approximation error implies order preservation, but **not vice versa** (Examples 1 and 3). **However**, determining how much larger the space is would require choosing a metric, such that we can compare space sizes. This needs further investigation.
>
> [1] Martin Grötschel, László Lovász, and Alexander Schrijver. Geometric algorithms and combinatorial
> optimization, volume 2. Springer Science & Business Media, 2012.

---

### Author Rebuttal · Authors · 2023-08-10

We thank the reviewers for their careful reading, valuable comments, and recognition of the contributions. This first, common feedback answers a question raised by multiple reviewers.

>**Generalization to MDPs (Reviewers m2vF, P4aQ, BpdK)**

Extending the results of this work to MDPs is an important and challenging next step as mentioned in the conclusion, and our work provides a new direction as the first step. Here we discuss some research plans, considering Softmax PG for illustration. The discussion provides some new ideas, but resolving this problem is highly non-trivial and requires further investigation.

According to the policy gradient theorem [21, Theorem 1], we have, for all $\theta_t \in \mathbb{R}^d$,
\begin{equation*}
    \theta_{t+1} = \theta_t + \eta \cdot \sum_{s \in \mathcal{S}} d^{\pi_{\theta_t}}(s) \sum_{a \in \mathcal{A}} \frac{\partial \pi_{\theta_t}(s, a)}{\partial \theta_t} Q^{\pi_{\theta_t}}(s, a)
\end{equation*}
\begin{equation*}
    = \theta_t + \eta \cdot \sum_{s \in \mathcal{S}} d^{\pi_{\theta_t}}(s) \cdot X_s^\top ( \text{diag}{(\pi_{\theta_t}(\cdot | s))} - \pi_{\theta_t}(\cdot | s)\pi_{\theta_t}(\cdot | s)^\top ) \ Q^{\pi_{\theta_t}}(s, \cdot),
\end{equation*}
where $X_s \in \mathbb{R}^{|\mathcal{A}| \times d}$ is the feature matrix under state $s \in \mathcal{S}$ and can be shared across multiple states. Comparing with Eq. (4), for all $s \in \mathcal{S}$, the reward vector $r \in \mathbb{R}^K$ is replaced by $Q^{\pi_{\theta_t}}(s, \cdot) \in \mathbb{R}^{|\mathcal{A}|}$, as mentioned also by Reviewer P4aQ. This fact provides some new ideas as well as difficulties.

**First**, if for all state $s \in \mathcal{S}$, the feature matrix can preserve the order of $Q^{\pi_{\theta_t}}(s, \cdot)$ for **all policies**, i.e., for all $t \ge 1$, there exists $w_t \in \mathbb{R}^d$, such that for all $s \in \mathcal{S}$, $X_s w_t \in \mathbb{R}^{|\mathcal{A}|}$ preserves the order of $Q^{\pi_{\theta_t}}(s, \cdot)$, then we have,
\begin{equation*}
    \theta_{t+1}^\top w_t = \theta_t^\top w_t + \eta \cdot \sum_{s \in \mathcal{S}} d^{\pi_{\theta_t}}(s) \cdot w_t^\top X_s^\top ( \text{diag}{(\pi_{\theta_t}(\cdot | s))} - \pi_{\theta_t}(\cdot | s)\pi_{\theta_t}(\cdot | s)^\top ) \ Q^{\pi_{\theta_t}}(s, \cdot)
\end{equation*}
\begin{equation*}
    \ge \theta_t^\top w_t,
\end{equation*}
generalizing Eq. (12), a key argument in the one-state setting. However, $w_t$ is  changing over time, since in the update $r \in \mathbb{R}^K$ is replaced by $Q^{\pi_{\theta_t}}(s, \cdot)$, which is changing over $\theta_t$. Comparing with fixed $r$ and $w$ in Eq. (12), using the above inequality with $\theta_t$ and $w_t$ both changing over time is more challenging and manifests the major technical difficulty.

**Second**, another speculation is that preserving the order of $Q^*(s, \cdot)$ (value of the optimal policy $\pi^*$) might be enough to achieve global convergence (if true, there would be no need to preserve the values of all policies). Here we show a local convergence when $\text{softmax}(X_s \theta_t)$ is close enough to $\pi^*( \cdot | s)$. Suppose that there exists $w^* \in \mathbb{R}^d$, such that for all $s \in \mathcal{S}$, $X_s w^* \in \mathbb{R}^{|\mathcal{A}|}$ preserves the order of $Q^*(s, \cdot)$. Then for any $\theta_t$ such that $Q^{\pi_{\theta_t}}(s, \cdot)$ preserves the order of $Q^*(s, \cdot)$, we have,
\begin{equation*}
    \theta_{t+1}^\top w^* = \theta_t^\top w^* + \eta \cdot \sum_{s \in \mathcal{S}} d^{\pi_{\theta_t}}(s) \cdot {w^*}^\top X_s^\top ( \text{diag}{(\pi_{\theta_t}(\cdot | s))} - \pi_{\theta_t}(\cdot | s)\pi_{\theta_t}(\cdot | s)^\top ) \ Q^{\pi_{\theta_t}}(s, \cdot)
\end{equation*}
\begin{equation*}
    \ge \theta_t^\top w^*,
\end{equation*}
based on which we can show that $\theta_t$ eventually approaches the direction of $w^*$, implying that $\pi_{\theta_t}(a^* | s) = \text{softmax}(X_s \theta_t)(a^*) \to \pi^*(a^* | s) = 1$ (in combination with Lemma 1). This means that preserving the order of $Q^*(s, \cdot)$ is enough for $\pi^*$ to be a local attractor of gradient updates within its neighbourhood. One challenge here is to generalize the arguments for arbitrary initialization $\theta_1 \in \mathbb{R}^d$ rather than $\theta_t$ close enough to optimal solution, and the difficulty is that $Q^{\pi_{\theta_t}}(s, \cdot)$ does not necessarily preserve the order of $Q^*(s, \cdot)$, and the last inequality above does not necessarily hold.

**In summary**, this discussion illustrates how the paper provides some new and useful insights for understanding more complex settings, but it requires further investigation to resolve this highly non-trivial problem for general MDPs. We will use an additional page in subsequent versions to present this discussion, as Reviewer P4aQ suggested.

---

### Decision · Program_Chairs · 2023-09-21

**Decision:**

Accept (oral)

**Comment:**

In this paper, the authors identify a new set of conditions that guarantee global convergence of policy gradient methods in the presence of linear function approximation. The results and the analysis strategy developed in this paper make solid contributions to this topic, which could help advance our understanding about the performance of this widely used policy optimization method. Note that the proof of the main results might require substantial writing in order to improve readability and clarity, which I hope the authors can do in the final version of this paper.